# The Pitfalls of Text Degeneration when Aligning LLMs through Model Merge

**Peijun Qing**  *peijun.qing.gr@dartmouth.edu*
*Dartmouth College*

**Hefan Zhang**  *hefan.zhang.gr@dartmouth.edu*
*Dartmouth College*

**Lei Hsiung**  *lei.hsiung.gr@dartmouth.edu*
*Dartmouth College*

**Haiquan Lu**  *haiquanlu@u.nus.edu*
*National University of Singapore*

**Xingjian Diao**  *xingjian.diao.gr@dartmouth.edu*
*Dartmouth College*

**Chiyu Ma**  *chiyu.ma.gr@dartmouth.edu*
*Dartmouth College*

**Saeed Hassanpour**  *saeed.hassanpour@dartmouth.edu*
*Dartmouth College*

**Soroush Vosoughi**  *soroush.vosoughi@dartmouth.edu*
*Dartmouth College*

**Reviewed on OpenReview:** *https://openreview.net/forum?id=zJAy9Tt9DX*

## Abstract

Model merge offers a cost-efficient method for integrating multiple specialized large language models (LLMs) into one comprehensive model. While it shows promise for encoder-decoder models in standard Natural Language Processing (NLP) tasks, **we find that merging decoder-based LLMs may lead to localized text degeneration, even when overall performance appears to improve.** We specifically assess the applications of model merge in steering LLMs to align better with diverse human preferences through interpolation and extrapolation merge. Our extensive experiments, covering model sizes ranging from 7b to 70b parameters, and including sixteen models with varying post-training, employ three popular merging methods: `Task Arithmetic`, `TIES-Merging`, and `Dare-TIES`. Our results uncover inherent limitations in current model merge applications for alignment, which can lead to text degeneration. We hope our findings will offer valuable insights for employing model merging in alignment scenarios and can help practitioners avoid potential pitfalls.

## 1 Introduction

Contemporary pre-trained language models (LMs) acquire instruction-following and conversational abilities largely through post-training techniques that operate on their next-token prediction objective (Ouyang et al., 2022; Bai et al., 2022; Brown et al., 2020; Xu et al., 2023; Yuan et al., 2025a; Qing et al., 2024; 2026b; Zhang et al., 2025a;b; Liu et al., 2024). Customizing LLMs for downstream applications typically relies on supervised fine-tuning (SFT) with human-written instructions (Ouyang et al., 2022) or preference-based optimization

such as Reinforcement Learning with Human Feedback (RLHF) (Bai et al., 2022) and Direct Preference Optimization (DPO) (Rafailov et al., 2024). Although these methods can substantially improve alignment with human preferences, they demand significant computational resources and costly annotation efforts (Bai et al., 2022).

Model merging has emerged as a simple and efficient alternative. Instead of training, it directly operates at the parameter level by combining the weights of multiple models, each endowed with different capabilities, to assemble a unified model without additional data or computation (Ilharco et al., 2022; Yang et al., 2024). Inspired by success in merging encoder-decoder models for broad capability gains, recent studies have begun applying merging techniques to decoder-only LLMs for alignment improvements (Zheng et al., 2024; Akiba et al., 2024; Yu et al., 2024). In particular, these works often merge models that have undergone different post-training procedures (e.g., SFT vs. RLHF/DPO) in an attempt to boost alignment without rerunning the full alignment pipeline (Figure 1).

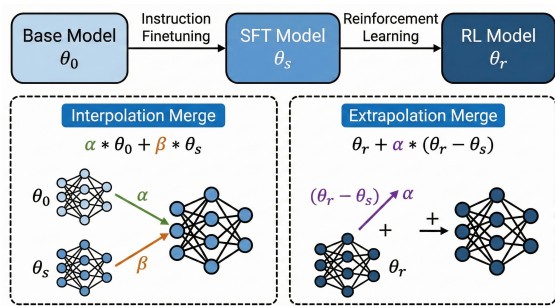

Figure 1: Illustration of contemporary post-training pipeline and two representative model merge settings.

At the same time, practitioners increasingly report that effective use of model merging resembles a "black art": merged models are often sensitive to seemingly minor choices of merge weights, target checkpoints, and sparsification schemes, and can exhibit surprising shifts in behavior (Akiba et al., 2024; Zheng et al., 2024; Yu et al., 2024). These shifts are particularly concerning in alignment settings. Post-training stages such as SFT and RLHF do not merely add "better" behavior; they imprint distinct inductive biases and failure modes on the underlying base model (Chu et al., 2025). As a result, merging models taken from different points in the post-training pipeline risks combining not only their strengths, but also their weaknesses. In particular, alignment training is known to incur an alignment tax: loss of pre-trained capabilities and side effects such as reduced diversity, language mixing, or brittle behavior off-distribution (Ouyang et al., 2022; Lin et al., 2024; Guo et al., 2025; Lu et al., 2024). When we treat post-training updates as task vectors and aggressively interpolate or extrapolate them, it remains unclear whether we are amplifying desirable alignment behavior, collapse-prone artifacts, or both.

To address this gap, we study failure modes of model merging when it is used as an alignment tool, and identify when merging behaves as benign capability composition versus when it triggers unstable generation. We evaluate merged models using standard preference-based metrics (reward models on instruction-following benchmarks) together with targeted diagnostics for localized text degeneration. We define degeneration as three observable behaviors—pathological repetition, uncontrolled code-switching, and nonsensical or garbled outputs—and use their prevalence to detect collapse-like instability.

Using this diagnostic, we systematically examine two representative uses of model merging for alignment: (1) extrapolating post-training updates (e.g., SFT→RL) back onto an RL-tuned model, and (2) interpolating between SFT/RL models derived from the same or different base checkpoints. Our study spans models from `7b` to `70b` parameters, covering sixteen open-source models with accessible base, SFT, and RL checkpoints, and three widely used merging algorithms: `Task Arithmetic` (Ilharco et al., 2022), `TIES-Merging` (Yadav et al., 2023), and `Dare-TIES` (Yu et al., 2024). This setup allows us to ask: when merging improves reward-model scores or downstream task performance, does it also quietly increase the prevalence of localized text degeneration?

Our findings highlight intrinsic constraints of current model-merging approaches for alignment: ① Both extrapolation and interpolation merges can induce text degeneration, even when standard performance improves. ② Advanced merging algorithms boost utility but still fail to suppress harmful spurious features that drive degeneration. ③ Interpolation merging is more reliable when combined models originate from the same checkpoint or share similar capabilities. ④ Merging models with shared domain characteristics substantially mitigates text degeneration. Taken together, these results suggest that model merging, while

attractive as a low-cost tool, must be applied with care: naive scaling or composition of post-training updates can push models toward collapse-like regimes, even when surface-level alignment metrics appear to improve.

## 2 Related Work

**Model Collapse and Alignment Tax.** Prior work has identified multiple mechanisms through which training procedures can alter a model's output distribution. *Model collapse* refers to a degenerative process arising from recursive self-training on synthetic data generated by earlier models (Shumailov et al., 2024; Dohmatob et al., 2024; Briesch et al., 2023). Repeated self-consuming training progressively erodes support for low-probability regions of the data distribution, leading to distributional shrinkage, loss of diversity, and irreversible degradation. Collapse is therefore a property of recursive training dynamics. A distinct but related phenomenon is the *alignment tax*, where post-training alignment methods such as RLHF or DPO shift model behavior toward human preferences at the expense of other capabilities (Ouyang et al., 2022; Lin et al., 2024). Empirical studies report trade-offs including reduced robustness, performance drops on unrelated benchmarks, narrowed behavioral diversity, and brittle off-distribution behavior (Lin et al., 2024; Lu et al., 2024; Guo et al., 2025; Wu et al., 2025a). Unlike model collapse, alignment tax does not require recursive synthetic data, but both phenomena involve training-induced shifts in output distributions. Our work differs from both lines of research. We do not study recursive self-training collapse, nor do we analyze alignment tax from a capability benchmarking perspective. Instead, we investigate instability that arises from weight-space merging of post-trained checkpoints. We observe collapse-like behavioral symptoms such as repetition and code-switching and aim to highlight that merging can push models toward regimes exhibiting the same surface-level degenerative behaviors identified in collapse studies.

**Model Merging.** Model merging combines multiple checkpoints directly in weight space, enabling capability composition without additional training or access to original data (Ilharco et al., 2022; Yadav et al., 2023; Yu et al., 2024; Yang et al., 2024; Yuan et al., 2026; 2025b). Existing approaches include simple weight averaging and interpolation, task-vector arithmetic, sparsification-based methods such as TIES and DARE, as well as data-aware or curvature-aware variants (e.g., Fisher-weighted merging). While these methods have demonstrated strong empirical gains across diverse benchmarks, most prior evaluations focus on aggregate task accuracy or utility. In contrast, our work examines merging from the perspective of generation stability and collapse-like degeneration. We provide a more detailed overview of model merging methods and related advances in Appendix D.

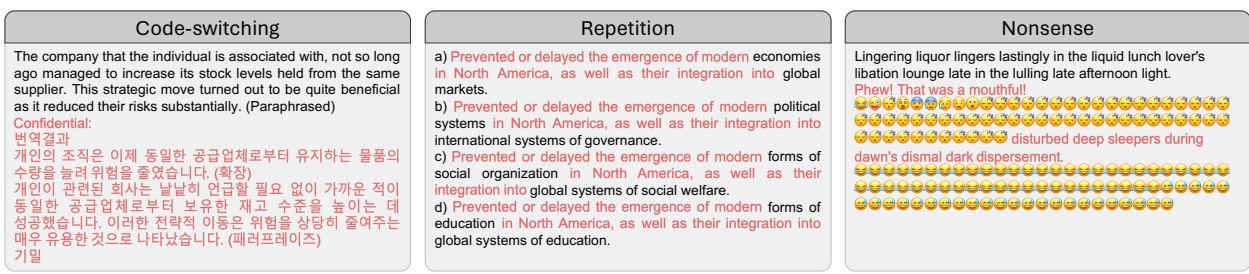

Figure 2: Example output of three different types of degeneration.

## 3 Text Degeneration

Text degeneration refers to pathological generation behaviors observable at inference time. Unlike model collapse or alignment tax—which describe training-time distributional shifts—degeneration is a *behavioral manifestation* in generated text. Degeneration has been extensively studied in neural text generation. Likelihood-trained models are known to produce bland or repetitive outputs under greedy decoding (Holtzman et al., 2019). Self-reinforcement effects during decoding can trap models in loops, leading to sentence-level repetition (Welleck et al., 2020; Xu et al., 2022). Data-level biases can further amplify repetition tendencies

(Li et al., 2023a). These works establish degeneration as a decoding- and distribution-sensitive behavioral failure mode.

We hypothesize that alignment-oriented model merging can amplify unstable or overly specialized parameter updates, increasing the frequency of degenerate outputs—even when aggregate reward scores improve. To operationalize degeneration, we categorize three observable failure types.

**Repetition.** Repetition is a well-documented generation failure. It may arise from decoding dynamics or training biases (Welleck et al., 2020; Xu et al., 2022). Repetition is not inherently undesirable—human discourse often uses repetition for emphasis. We classify repetition as *degenerate* when the model produces near-identical tokens, phrases, or sentences in a self-looping manner without semantic progression.

**Code-switching.** Code-switching refers to alternating between languages within discourse (Sitaram et al., 2019). In multilingual contexts this is natural. However, unintended language mixing has been observed after certain alignment procedures (Lu et al., 2024; Guo et al., 2025). We define degenerate code-switching as unexpected language or format shifts not prompted by the user, which degrade readability or coherence.

**Nonsense.** Nonsense denotes output segments that are irrational, syntactically malformed, semantically incoherent, or unrelated to the instruction. Such behavior has been associated with distributional drift, decoding instability, or data contamination (Shumailov et al., 2024; Briesch et al., 2023). In our setting, nonsense reflects inference-time instability rather than mere task inaccuracy.

These categories serve as diagnostic indicators of instability in open-ended generation. Degeneration differs from ordinary low-quality answers (e.g., factually incorrect but well-formed responses); it refers to patho-logical looping, incoherence, or formatting breakdown that impairs interpretability. Throughout the paper, degeneration rates are used as a behavioral signal of merge-induced instability.

## 4 Model Merging Methods

**Notation** As shown in Figure 1, given a set of pre-trained base models $\boldsymbol{\theta}_o^1, \ldots, \boldsymbol{\theta}_o^N$, we denote their corresponding SFT models which undergo supervised instruction fine-tuning as $\boldsymbol{\theta}_s^1, \ldots, \boldsymbol{\theta}_s^N$. Similarly, the RL model is the SFT model that goes through reinforcement learning (i.e., DPO or RLHF), denoted as $\boldsymbol{\theta}_r^1, \ldots, \boldsymbol{\theta}_r^N$. The weight change from base model to SFT model and from SFT to RL model are denoted as $\Delta\boldsymbol{\theta}_s$ and $\Delta\boldsymbol{\theta}_r$, respectively.

We explore three popular merging techniques: Task Arithmetic (Ilharco et al., 2022), TIES-Merging (Yadav et al., 2023), and DARE (Yu et al., 2024). These methods exhibit outstanding performance in encoder-decoder architectures and show adaptability to larger decoder-based models. We exclude more complex approaches, such as those requiring Fisher matrix computations (Matena & Raffel, 2022), backward passes (Yang et al., 2023), or additional data like model activations (Jin et al., 2023), due to their high computational demands in large-scale model merging. Subsequently, we provide a detailed exploration of the three selected techniques.

### 4.1 Task Arithmetic

In task arithmetic (Ilharco et al., 2022), a pre-trained model with parameters $\theta_{\text{pre}} \in \mathbb{R}^d$ is fine-tuned on a task $t$, resulting in parameters $\theta_t \in \mathbb{R}^d$. The difference between the two models, $\Delta\boldsymbol{\theta}_t = \theta_t - \theta_{\text{pre}}$, is termed a task vector. These vectors can be manipulated using operations like addition and negation to edit the model's parameters for specific tasks or to accommodate multiple tasks. For a set of tasks $T$, task vectors are combined and scaled by empirical factors $\lambda_t$ to form $\theta_{\text{new}} = \theta_{\text{pre}} + \sum_{t \in T} \lambda_t \Delta\boldsymbol{\theta}_t$, effectively allowing flexible model adjustment. We mainly employ this merging method as different merging methods perform similarly when applied to large-scale instruction-tuned models (Yu et al., 2024; Yadav et al., 2024).

## 4.2 `TIES` **Merging**

`TIES`-Merging (Yadav et al., 2023) identifies two main challenges with model merging: 1) during fine-tuning, expert models accumulate noise in the parameters, and 2) different experts might want to change the same parameter in different directions, leading to interference or conflict. A task vector, $\Delta\boldsymbol{\theta}$, contains both the direction and the movement scale needed for optimal task performance. Each element in the task vector represents an axis guiding how to adjust parameters to decrease task loss. The vector can be expressed as $\Delta\boldsymbol{\theta} = \gamma \odot \mu_{\Delta\boldsymbol{\theta}}$, where $\gamma = \mathrm{sgn}(\Delta\boldsymbol{\theta})$ encodes direction, and $\mu_{\Delta\boldsymbol{\theta}} = |\Delta\boldsymbol{\theta}|$ provides magnitude. To merge multiple task models $\{\theta_t\}_{t=1}^n$, we convert them to task vectors $\{\Delta\boldsymbol{\theta}_t\}_{t=1}^n$ and follow these steps:

1) **Trim:** For each task $t$, remove *less significant parameters* in $\Delta\boldsymbol{\theta}_t$ to form $\hat{\Delta\boldsymbol{\theta}}_t$, i.e., keeping the top-$k\%$ based on magnitude and setting the rest to zero.

2) **Elect:** Calculate an *aggregate* sign vector $\gamma_m$, which handles sign conflicts parameter-wise, choosing the sign of each parameter based on the highest magnitude sum. Therefore, the sign of each parameter $p$ is $\gamma_m^p = \mathrm{sgn}(\sum_{t=1}^n \hat{\Delta\boldsymbol{\theta}}_t^p)$.

3) **Disjoint Merge:** For each parameter, calculate a mean only from models where signs align with $\gamma_m$. Let $\mathcal{A}^p = \{t \mid \hat{\gamma}_t^p = \gamma_m^p\}$, then the parameter of the final task vector is $\Delta\boldsymbol{\theta}_m^p = \frac{1}{|\mathcal{A}^p|} \sum_{t \in \mathcal{A}^p} \hat{\Delta\boldsymbol{\theta}}_t^p$.

## 4.3 DARE Merging

Building upon `TIES` merging, DARE (Yu et al., 2024) incorporates a preliminary pruning stage inspired by dropout techniques. This stage aims to remove noise from task vectors prior to merging. Specifically, each task vector $\Delta\theta$ undergoes a stochastic pruning process where elements are randomly zeroed out. This is achieved by applying a Bernoulli mask $\mathbf{m}$, whose elements are 1 with a drop probability $p$. The expected value of the task vector is preserved by rescaling the non-pruned elements. The resulting pruned vectors are then suitable for use with either `TIES` Merging or Task Arithmetic. The widely adopted Dare-TIES variant, which employs `TIES` Merging, defines this pruning and rescaling as:

$$\mathbf{m} \sim \mathrm{Bernoulli}(p), \tag{1}$$

$$\tilde{\Delta}\theta = (1 - \mathbf{m}) \odot \Delta\theta, \tag{2}$$

$$\hat{\Delta}\theta = \tilde{\Delta}\theta / (1 - p). \tag{3}$$

In this formulation (Eq. 1–3), each parameter in $\Delta\theta$ is set to zero with probability $p$. The term $(1 - \mathbf{m})$ in Eq. 2 effectuates this, and the subsequent division by $(1 - p)$ in Eq. 3 ensures the expected value remains unchanged. Yu et al. (2024) show that many parameters in $\Delta\theta$ are redundant, allowing for their removal via this method without a critical loss in task-specific model performance.

# 5 Experimental Results

In this section, we examine two representative merging scenarios to enhance alignment: (1) extrapolating the $\Delta\boldsymbol{\theta}$ learned during post-training back to the RL models (Section 5.2 and 5.3), and (2) interpolating different SFT/RL models derived from varied or identical base models (Section 5.4 and 5.5).

## 5.1 Experimental Setting

We assess alignment with human preferences using reward scores from a strong open-source reward model on Reward-Bench (see Appendix A), following the evaluation protocol

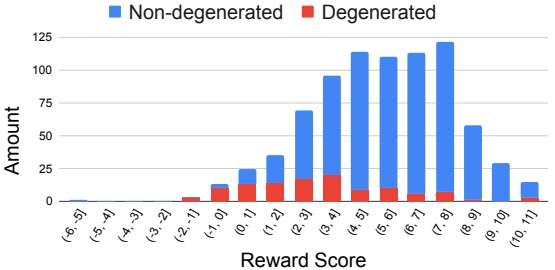

Figure 3: Distribution of reward-model scores on AlpacaEval 2.0 for responses labeled as degenerated (red) and non-degenerated (blue).

of ExPO (Zheng et al., 2024). To study collapse-like be-
havior, we augment this scalar score with a diagnostic of
*text degeneration.* As a first step, we empirically verify that degeneration is associated with low reward
scores. We run models on AlpacaEval 2.0 (Li et al., 2023b), score the responses with the reward model, and
use GPT-4 (LLM-as-a-Judge) to annotate whether each response exhibits any degeneration type defined
in Section 3 (repetition, code-switching, or nonsense). Figure 3 shows the distribution of reward scores for
degenerated versus non-degenerated responses. Degenerated outputs are concentrated in the lower-score bins
and are rare for high scores, indicating that the reward model is sensitive to overt degeneration and that low
scores provide a good proxy for severe failures. Motivated by this correlation, our main evaluation isolates
the worst-performing generations for each model and merge configuration. Concretely, for each configuration
on AlpacaEval 2.0, we score all responses with the reward model. Then we select the 100 responses with
the lowest reward scores. Finally, we prompt GPT-4 to determine whether each selected response exhibits
repetition, code-switching, or nonsense. Any response that displays at least one of these traits is marked
as *degenerate.* The *degeneration rate* of a model is then defined as the fraction of these 100 low-reward
responses that are labeled as degenerate. This rate serves as a qualitative indicator of collapse-like behavior,
complementary to aggregate reward scores and downstream task metrics. We conduct additional experiments
to validate the robustness of our evaluation protocol. Specifically, we (1) replicate extrapolation results using
different open-source reward model; (2) evaluate degeneration using multiple judge models; (3) vary the
tail threshold used to compute degeneration rate (Top-50, 100, 150, 200); and (4) test different decoding
temperatures. Detailed results are reported in Appendix E.

**Models**  To ensure transparent experimental analyses, we select open-source LLMs that provide access
to their base models, corresponding SFT models, and RL models. Notably, many popular models like
LLaMA-2/3 (Touvron et al., 2023; Dubey et al., 2024) and Gemma (Team et al., 2024) only release their base
models and final RL models, withholding the intermediary SFT models. Thus, we select models releasing
all checkpoints from various post-training stages, with each comprehensively available on HuggingFace (see
Appendix A for a detailed list of models).

**Implementation**  We utilize the `vllm` library (Kwon et al., 2023), which facilitates high-throughput
inference across all models. We employ top-$k$ sampling with $k = 50$, combined with nucleus sampling at
$p = 0.9$, and a temperature setting of 0.7. To mitigate repetitive content in the generated text, both presence
and frequency penalties are set at 0.1. We maintain consistent decoding hyperparameters throughout all
experiments, deploying a sampling random seed of 42 for all models evaluated. All experiments are conducted
with eight RTX A6000-48G GPUs.

## 5.2 Extrapolation from SFT to RL Model

In this section, we investigate the effect of incremental increases in the extrapolation coefficient.

**Experimental Setting**  Following the ExPO framework (Zheng et al., 2024), we consider the extrapolation
from SFT models to RL models, as it effectively enhances performance, unlike the progression from base
models to SFT models. We selected six open-source models with publicly accessible SFT and RL versions:
Zephyr-7b-alpha (Tunstall et al., 2023), Starling-7b-alpha (Zhu et al., 2024), Llama3-8b-iter (Dubey et al.,
2024), and Tulu-2-dpo-7/13/70b (Ivison et al., 2023). The task vector $\Delta\boldsymbol{\theta}$ is computed by subtracting SFT
weights from RL weights using the task arithmetic merging described in Section 4.1. This task vector is then
added back to the RL model with different scaling factors of $\{0.3, 0.5, 0.8\}$, as recommended by ExPO (Zheng
et al., 2024). Further details about all models are provided in Appendix A.

**Findings**  Figure 4 (Left) illustrates that increasing the weight of the reward vector from 0.3 to 0.5 results
in linear improvements in reward scores for five out of six models, at the cost of rising degeneration rate.
Increasing the weight to 0.8 causes a decline in performance, likely due to the amplification of unintended
features in the reward vector, as discussed by ExPo (Zheng et al., 2024). Across all six models, improvements
in performance are accompanied by increased degeneration rate, even when overall performance appears to
improve. This highlights the trade-off between performance and stability inherent in extrapolation merging.

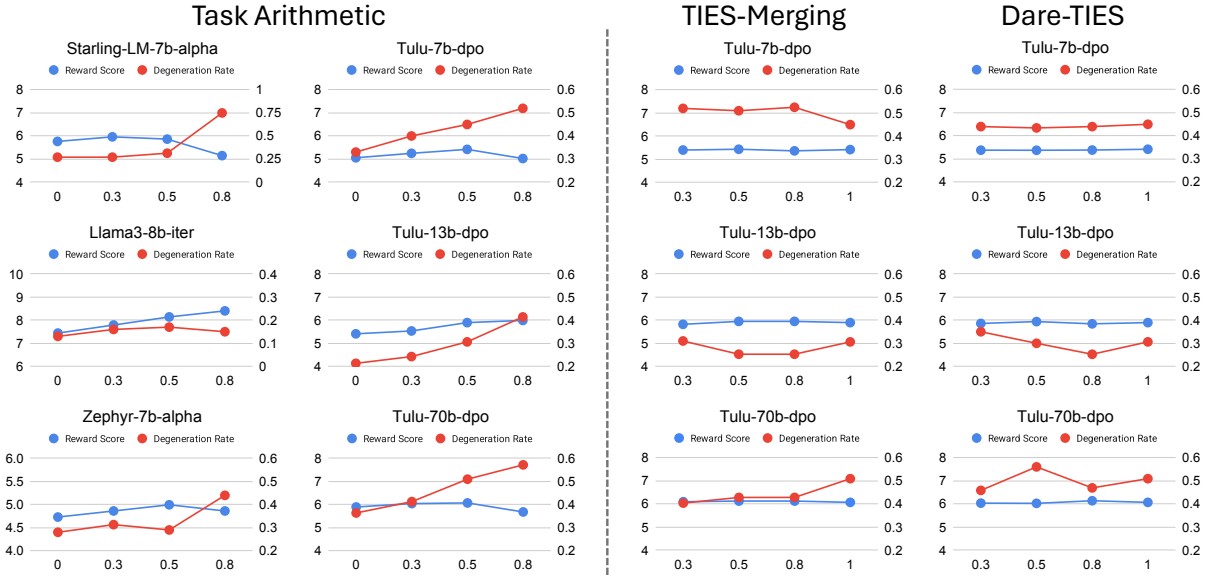

Figure 4: The reward score (left y-axis) and the degeneration rate (right y-axis) of different models and merging methods. **(Left)** For task arithmetic merging, the x-axis is the coefficient used for adding the task vector. **(Right)** For `TIES-Merging` and `Dare-TIES`, the x-axis is the density parameter.

### 5.3 Retaining Influential Parameters May Not Prevent Degeneration

In this section, we investigate whether advanced merging methods that retain the influential parameters can boost performance and mitigate degeneration.

**Experimental Setting** We select three merging configurations from Section 5.2 of Tulu-2 series models, employing both `TIES-Merging` and `Dare-TIES` methods. Specifically, we fix the scaling factor to 0.5, where the overall reward score is optimized while the degeneration rate increases. We examine the density parameter within $\{0.3, 0.5, 0.8, 1\}$, where 1 is the initial task arithmetic merging in Section 5.2. The density parameter serves to alleviate interference between parameters of different models by retaining top-$k\%$ most influential parameters and zeroing out others. Detailed descriptions of these advanced merging techniques can be found in Section 4.2 and 4.3.

**Findings** As illustrated in Figure 4 (Right), we observe that: ① **Adjusting density does not significantly affect the overall reward score**, indicating that the delta parameter (i.e., the difference between aligned and pre-aligned parameters) acquired during DPO/RLHF training demonstrates redundancy, akin to the delta parameter obtained by pre-trained language models following SFT. ② **Retaining only influential parameter values does not necessarily prevent degeneration.** In the tulu-7b-DPO model, neither merging method reduces the degeneration rate across different densities. For larger models, setting the density to 0.5 or 0.8 can alleviate redundancy. We conjecture this is because, in larger models, a lower density may zero out more parameters, thus eliminating spurious features and subsequently reducing the degeneration rate. However, it is challenging to accurately distinguish significant parameters contributing to instruction compliance while excluding features causing degeneration, as shown by the fluctuating degeneration rate with decreasing density.

### 5.4 Interpolation between SFT or RL Models Results in In-Between Performance

We next examine whether interpolation merging can successfully integrate capabilities across instruction-tuned or RL-tuned models. Unlike extrapolation, which amplifies a single post-training update, interpolation merges two aligned models by linearly scaling and combining their respective task vectors. Prior work has suggested that interpolation may yield in-between behavior (Zheng et al., 2024), but the stability and alignment

consequences of these merges remain underexplored—particularly when the source models originate from different base checkpoints or have undergone divergent post-training pipelines. We experiment with the following combinations:

(1) **SFT+SFT - 1** (same base): Zephyr-alpha-SFT and Zephyr-beta-SFT (Tunstall et al., 2023), two SFT models initialized from Mistral-7b-v0.1 (Jiang et al., 2023) developed by HuggingFace.

(2) **SFT+SFT - 2** (different base): Mistral-7b-Instruct-v0.1 and Mistral-7b-Instruct-v0.2 (Jiang et al., 2023), two SFT models initialized from Mistral-7b-v0.1 and Mistral-7b-v0.2, respectively.

(3) **RL + RL - 1** (same base): Zephyr-7b-alpha and Zephyr-7b-beta (Tunstall et al., 2023), two RL models initialized from the same base model (Mistral-7b-v0.1) but different SFT models.

(4) **RL + RL - 2** (same SFT): Snorkel-7b (Tran et al., 2023) and Mistral-7b-SPPO (Wu et al., 2025b), two RL models initialized from the same SFT model (Mistral-7b-Instruct-v0.2).

(5) **RL + RL - 3** (different base): Snorkel-7b (Tran et al., 2023) and Zephyr-7b-beta, two RL models initialized from Mistral-7b-v0.2 and Mistral-7b-v0.1, respectively.

For each pair, we sweep interpolation coefficients $\lambda \in \{0, 0.2, 0.4, 0.6, 0.8, 1\}$ and evaluate reward model performance on AlpacaEval 2.0.

**Findings** Figure 5 shows that simple linear interpolation between SFT or RL models almost always yields *in-between* behavior. For all five configurations, the reward score of the merged model lies within the convex hull of the two endpoints, and the curves are approximately linear in the interpolation coefficient. In particular, we do not observe any consistent regime in which naive interpolation strictly dominates both constituent models. This suggests that directly mixing the $\Delta\boldsymbol{\theta}$

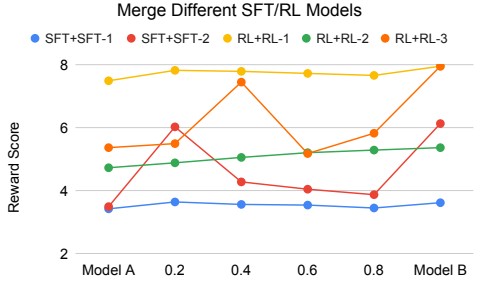

Figure 5: The left axis is reward score. The x-axis is the interpolation coefficients.

updates learned during SFT or RL largely averages their effects rather than composing complementary capabilities in a synergistic way. The **RL+RL-3** pair, which interpolates models trained from different base checkpoints, is especially unstable: its performance remains close to or below that of the weaker endpoint across all coefficients, indicating that mismatch in pre-training or post-training pipelines can make interpolation merging ineffective or even harmful.

Figure 6 examines the same model pairs under the advanced `TIES-Merging` and `Dare-TIES` merging schemes, which sparsify and reconcile task vectors before interpolation. For **SFT+SFT-1** (same base) and **RL+RL-2** (same SFT), we observe that, across a broad range of interpolation coefficients and density values, both `TIES-Merging` and `Dare-TIES` can match or slightly exceed the reward score of the better individual model. This indicates that when models share a common initialization or SFT checkpoint, selectively retaining and aggregating influential parameters can produce mild but consistent gains over naive averaging. In contrast, for **RL+RL-1**, where the two RL models are initialized from different SFT checkpoints, the benefits of `TIES-Merging` and `Dare-TIES` are weaker and less stable across densities, and for cross-base pairs (e.g., **RL+RL-3**) we do not observe systematic improvements.

Overall, these results support two conclusions. First, interpolation in parameter space behaves largely as expected from a linear model: it produces smooth, in-between performance without reliably unlocking new capabilities. Second, more sophisticated merging procedures can offer incremental improvements, but primarily when the source models are closely related (same base or same SFT). When post-training trajectories diverge too much, even advanced methods struggle to reconcile their updates into a consistently stronger merged model.

## 5.5 Impact of Expert Model Integration on Degeneration

**Experimental Setting** We select three decoder-based models, all derived from `Llama-2-13b` (Touvron et al., 2023) for different specialists. WizardLM (Xu et al., 2023) serves as the foundational instruction-

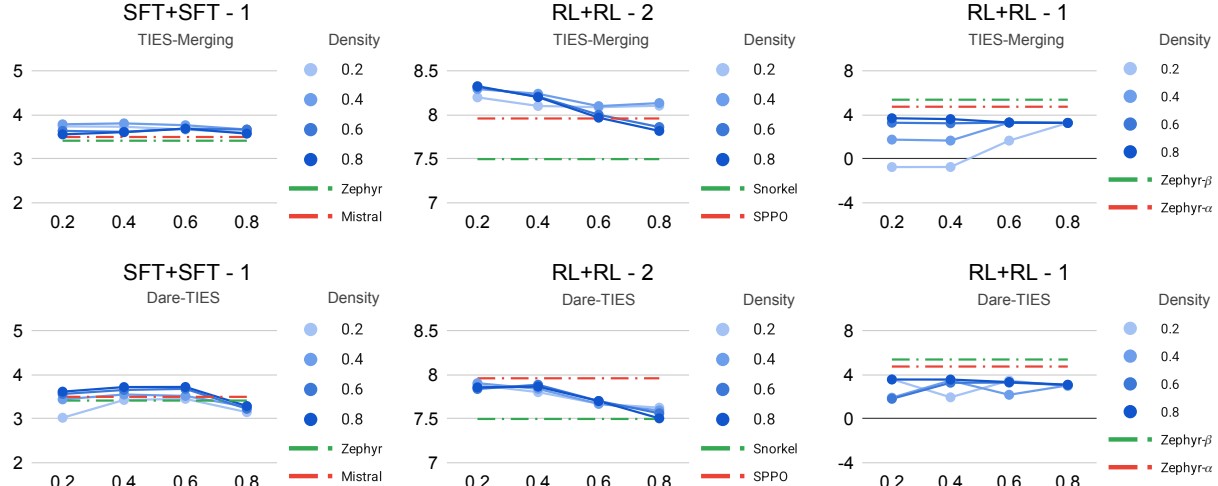

Figure 6: The left y-axis is reward score, and the right is density parameter of advanced merging methods in Section 4. The x-axis is the interpolation coefficients.

following model, referred to as Chat. For domain-specific capabilities, we employ `MetaMath-13B-V1.0`(Yu et al., 2023) as the mathematical expert model and `Code-13B`[1] as the coding expert model, denoted as Math and Code, respectively. We evaluate mathematical ability on GSM8K (Cobbe et al., 2021), and code-generation capability on MBPP (Austin et al., 2021).

**Findings** Figure 7 reveals a consistent trade-off between domain specialization and stability when integrating expert models into the chat model. For Chat+Math (left), increasing the Math coefficient substantially improves mathematical performance on GSM8K once the coefficient exceeds 0.4, but this comes at a clear cost: the reward score decreases almost monotonically, and the degeneration rate rises sharply from ≈ 0.18 at coefficient 0 to nearly 0.9 at 0.8. Thus, the math expert update behaves like a strongly domain-specific task vector that boosts reasoning in its target domain while simultaneously introducing collapse-like behavior and reducing general alignment quality. In contrast, Chat+Code (middle) exhibits a much milder trade-off. As the Code coefficient increases, coding performance on MBPP improves steadily, while the reward score stays within a relatively narrow band and only drops noticeably at the largest coefficient. The degeneration rate remains low and roughly flat for moderate coefficients and only spikes at 0.8. This pattern suggests that the chat and code experts are more compatible—likely sharing training data characteristics or instruction-following style—so that their task vectors can be merged without strongly destabilizing generation.

The Chat+Math+Code configuration (right) further illustrates how combining multiple experts compounds these dynamics. As the joint expert coefficient grows, both GSM8K and MBPP performance improve, but the reward score decreases and the degeneration rate increases from about 0.18 to 0.58. The trajectory closely resembles the behavior observed in Chat+Math, indicating that the math expert dominates the instability introduced by multi-expert merging.

Overall, these results show that expert integration is not free: substantial domain gains, especially in mathematics, are tightly coupled with increased degeneration and reduced general reward. Merging experts that share similar training pipelines or domains (e.g., chat and code) leads to more favorable trade-offs than merging experts whose updates are more divergent from the base chat model.

# 6 Discussion, Takeaways, and Future Work

If not applied with care, model merging can amplify collapse-like behaviors, leading to unstable decision-making, reduced robustness, and degraded user experience, even when standard metrics suggest improvements.

---
[1] `https://huggingface.co/ajibawa-2023/Code-13B`

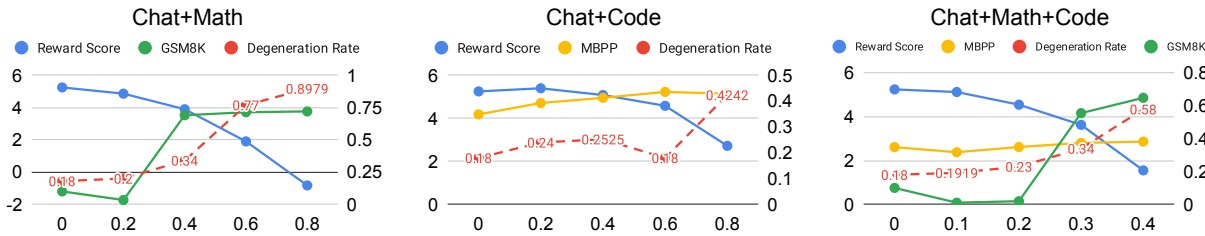

Figure 7: The performance variation of the merged model as the coefficient of the expert model is incrementally increased. The x-axis represents the coefficient applied to integrate the task vector.

Our study provides an initial, systematic look at these risks in alignment-focused merging setups. Below, we summarize key takeaways and outline directions for future work.

① **Extrapolating task vectors trades off alignment scores against degeneration.** Extrapolating post-training updates (e.g., from SFT to RL) with larger scaling coefficients generally yields near-linear gains in reward-model scores, but consistently increases the degeneration rate. This suggests that extrapolation amplifies not only desirable preference-aligned behavior, but also spurious or misaligned features encoded in the same update direction.

② **Sparsifying or denoising parameter updates is not sufficient to prevent collapse-like behavior.** Advanced merging schemes such as `TIES-Merging` and `Dare-TIES` confirm that post-training deltas are highly redundant: substantial sparsification has only a mild effect on reward scores. However, selectively retaining "influential" parameters does *not* reliably suppress degeneration. In several settings, degeneration remains elevated or even fluctuates non-monotonically as density decreases, indicating that influential parameters for instruction-following and those responsible for degeneration are intertwined and hard to disentangle at the level of simple magnitude- or sign-based heuristics.

③ **Interpolation mostly yields in-between behavior, with limited upside and non-trivial risk.** Linear interpolation between SFT or RL models produces performance that lies between the two endpoints, with little evidence of emergent capabilities. More sophisticated interpolation via `TIES-Merging`/`Dare-TIES` can occasionally outperform both constituent models, but the gains are modest and largely restricted to cases where models share a common base checkpoint or SFT initialization. In line with our extrapolation results, these improvements can still be accompanied by higher degeneration rates, suggesting that "stronger" merged models may be more brittle than either parent.

④ **Merging models with shared training or domain characteristics is empirically safer.** Merging models that share substantial training pipeline overlap (e.g., same base model, same SFT, or closely related domains) yields more stable trade-offs between utility and degeneration. In our expert-integration experiments, incorporating a coding expert into a chat model produces smoother gains and relatively modest increases in degeneration, whereas merging a math expert leads to large capability improvements but sharp increases in degeneration. This highlights that compatibility of post-training trajectories is a key, but currently under-theorized, ingredient for successful merging.

Looking forward, several research directions emerge from our findings. First, developing adaptive scaling strategies for task vectors—potentially conditioned on prompts or equipped with regularizers that explicitly suppress collapse-prone behaviors such as repetition or uncontrolled code-switching—may offer safer alternatives to fixed global coefficients. Second, a finer-grained mechanistic analysis of post-training deltas could help identify which components of $\Delta\theta$ encode alignment benefits versus degeneration drivers, enabling more structured merging at the level of layers, modules, or representation subspaces. Third, there is a need for more comprehensive, merge-aware evaluation protocols that extend beyond our simple diagnostic to encompass multilingual assessments, safety audits, and long-context robustness tests. Finally, applying similar analyses to proprietary models and more diverse alignment pipelines—including multi-stage RL, tool-augmented systems, and online feedback loops—would clarify how broadly the observed merge instabilities generalize and whether certain training regimes yield inherently more merge-compatible model families.

## 7 Limitations

Our study has several limitations that should be kept in mind when interpreting the results.

First, we restrict our analysis to open-source decoder-based LLMs for which base, SFT, and RL checkpoints are all publicly available. This excludes many widely deployed systems (e.g., LLaMA-2/3 and Gemma variants without released SFT checkpoints) and virtually all closed-source commercial models. The observed behaviors may differ in scale or character for models trained with larger data, more extensive safety pipelines, or proprietary infrastructure. Second, our evaluation heavily relies on automatic metrics. Alignment quality is measured using a single strong open-source reward model on AlpacaEval 2.0, and degeneration is diagnosed by applying GPT-4 as an LLM-as-a-judge to a small subset of low-reward responses. While prior work suggests that such evaluators correlate reasonably with human judgments, they are imperfect proxies and may miss subtler forms of failure or overestimate the severity of others. A more comprehensive human evaluation would be needed to fully validate our conclusions.Finally, all experiments are conducted under a fixed decoding recipe and a fixed random seed. Different decoding strategies or sampling temperatures could attenuate or exacerbate degeneration, and our conclusions about collapse-like behavior should be understood as conditional on this common decoding setup.

Despite these limitations, we believe our study provides a useful first step toward understanding the risks of model merging for alignment and motivates more systematic, human-centered evaluation of merged LLMs in future work.

## 8 Conclusion

In this paper, we systematically evaluate the degeneration issue caused by model merging in aligning LLMs with human preferences. Despite its cost-efficiency, merging models, particularly decoder-based ones, can lead to model collapse. This is indicated by increasing numbers of degenerated responses. Our experiments highlight the unpredictable nature of merged models, emphasizing the need for more robust merging techniques.

## 9 Acknowledgements

This research was supported in part by the National Library of Medicine (R01LM013833).

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

## A   Details of Open-sourced Models in Our Study

| Base models | SFT models | RL models |
|---|---|---|
| mistral-7b-v0.1 | mistral-7b-sft-alpha | zephyr-7b-alpha |
| mistral-7b-v0.1 | mistral-7b-sft-beta | zephyr-7b-beta |
| mistral-7b-v0.1 | zephyr-7b-sft | zephyr-7b-dpo |
| mistral-7b-v0.1 | openchat-3.5 | starling-7b-alpha |
| llama-3-8B | llama3-8b-sft | llama3-8b-iter |
| llama-2-7/13/70b | tulu-2-7/13/70b | tulu-2-dpo-7/13/70b |
| mistral-7b-v0.1 | mistral-7b-instruct-v0.1 | starling-7b-alpha |
| mistral-7b-v0.2 | mistral-7b-instruct-v0.2 | mistral-7b-SPPO |
| mistral-7b-v0.2 | mistral-7b-instruct-v0.2 | snorkel-7b |

Table 1: Open-source models used in our experiments.

| Models | HuggingFace ID |
|---|---|
| reward model | weqweasdas/RM-Mistral-7B |
| mistral-7b-sft-alpha | HuggingFaceH4/mistral-7b-sft-alpha |
| zephyr-7b-alpha | HuggingFaceH4/zephyr-7b-alpha |
| mistral-7b-sft-beta | HuggingFaceH4/mistral-7b-sft-beta |
| zephyr-7b-beta | HuggingFaceH4/zephyr-7b-beta |
| openchat-3.5 | openchat/openchat-3.5 |
| starling-7b-alpha | berkeley-nest/Starling-LM-7B-alpha |
| openchat-3.5-0106 | openchat/openchat-3.5-0106 |
| starling-7b-beta | Nexusflow/Starling-LM-7B-beta |
| mistral-7b-instruct-v0.2 | mistralai/Mistral-7B-Instruct-v0.2 |
| snorkel-7b | snorkelai/Snorkel-Mistral-PairRM-DPO |
| llama3-8b-sft | RLHFlow/LLaMA3-SFT |
| llama3-8b-iter | RLHFlow/LLaMA3-iterative-DPO-final |
| tulu-2-7b | allenai/tulu-2-7b |
| tulu-2-dpo-7b | allenai/tulu-2-dpo-7b |
| tulu-2-13b | allenai/tulu-2-13b |
| tulu-2-dpo-13b | allenai/tulu-2-dpo-13b |
| tulu-2-70b | allenai/tulu-2-70b |
| tulu-2-dpo-70b | allenai/tulu-2-dpo-70b |
| zephyr-7b-sft | alignment-handbook/zephyr-7b-sft-full |
| zephyr-7b-dpo | alignment-handbook/zephyr-7b-dpo-full |
| mistral-7b-v0.1 | mistralai/Mistral-7B-v0.1 |
| mistral-7b-instruct-v0.1 | mistralai/Mistral-7B-Instruct-v0.1 |

Table 2: Open-source models and HuggingFace IDs.

## B   Prompt

## C   Potential Applications of Model Merging

Beyond alignment-oriented checkpoint composition, model merging may provide a lightweight mechanism for integrating specialized capabilities in retrieval, document intelligence, RAG, information extraction, long-context reasoning, and code intelligence. Recent work on multimodal document systems highlights the need for heterogeneous skills such as layout-aware retrieval, multimodal parsing, evidence selection, and document-level reasoning (Dong, 2025; Dong et al., 2025b;a; Dong, 2024). Merging could combine checkpoints specialized

```
Evaluate the following generated response for any signs of degeneration.

Degeneration types:

A. Repetition: Does the response include phrases repeated in a way that
doesn't enhance the logic, or are there sentences stuck in repetitive
loops lacking new information?

B. Code-Switching: Does the response switch between several languages in a
way that disrupts the flow or meaning?

C. Nonsense: Are there portions of the response that appear nonsensical,
such as random incoherent gibberish, garbled characters, or Mojibake?

Instruction and Response:
"{}"

Based on your evaluation, respond with either 'Yes' or 'No' to indicate if
degeneration is present. Then, identify the letter labels A, B, or C,
which specify the type of degeneration and the reason.
Please provide your evaluation results in the following json format by
filling in the placeholders in []:

{{
"is_degeneration": "['Yes' or 'No']",
"type": "[select from options A, B, or C, stored as a list]",
"reason": "[explanation for each identified type in the response, stored
as a list]"
}}
```

Figure 8: Complete prompt for instructing GPT-4 to identify degenerated responses.

for visual retrieval, OCR/text retrieval, layout parsing, and multimodal generation without retraining a unified system. Similarly, adaptive and agentic RAG systems require retrieval control, reasoning, and answer synthesis, making model merging a practical way to compose complementary capabilities (Huanshuo, 2025; Zhang et al., 2025f). Related opportunities appear in prompt optimization, memory abstraction, structured knowledge extraction, and logical reasoning over knowledge graphs (Shi et al., 2024; Qing et al., 2026a; Dong, 2021; 2023; Yang et al.). Code intelligence is another promising setting, since repository-level assistants increasingly require both retrieval and generation abilities (Li, 2025).

Model merging may also support efficient inference, multimodal perception, long-term memory, personalization, and scientific reasoning. Recent pruning and token-reduction methods suggest that specialized computation can often be localized or compressed (Zhang et al., 2025c;d), motivating the composition of compact specialists rather than training monolithic models. Related multimodal work explores expert adapters, multimodal generation, visual quality assessment, temporal memory, and explainable visual reasoning (Wang et al., 2025; Xie et al., 2025; 2026; Liu et al., 2025; Xiao et al., 2025; Diao et al., 2025c;a). These directions suggest that merging could integrate models specialized for temporal reasoning, visual grounding, multimodal consistency, and memory management. Long-term memory systems further highlight the need to combine temporal,

personal, and multimodal representations over extended horizons (Wang et al., 2026; Hu et al., 2026a;b; Diao et al., 2025c).

Scientific and biomedical AI provide additional opportunities for model merging. Domain-specific multi-modal systems, including language-aligned fMRI understanding, medical imaging, pathology prediction, and MRI-based diagnosis, often require combining specialized perception with general reasoning and language generation (Wei et al., 2025; Qi et al., 2025a;b; Luo et al., 2025; Cong et al., 2025). Merging could integrate experts trained on different modalities, prediction targets, or institutional distributions without centralized retraining. Similar opportunities exist in audio-language reasoning, where specialized post-training improves reasoning over audio-text inputs (Diao et al., 2025b). However, representation compatibility remains critical, since forcing heterogeneous representations together may reduce robustness (Zhang et al., 2025e).

Finally, these applications require evaluation beyond average task performance. Interactive systems must handle ambiguous intent, uncertain contexts, and human feedback (He et al., 2025a), while LLM-as-Judge studies show that collaborative evaluation can amplify biases such as verbosity, position, and bandwagon effects (Ma et al., 2025; Zhou et al., 2025). Our results therefore suggest that merged systems should be evaluated not only on accuracy or reward, but also on hallucination, degeneration, structured-output reliability, cross-modal consistency, memory failures, and robustness under difficult inputs.

## D   Extended Discussion on Model Merging

Model merging combines multiple checkpoints into a single model directly in parameter space, avoiding additional gradient updates or access to the original training data (Akiba et al., 2024; Ilharco et al., 2022; Yang et al., 2024; Yadav et al., 2024; Utans, 1996; Yadav et al., 2023). The simplest form is weight averaging, which averages multiple fine-tuned models and can improve accuracy and robustness when the checkpoints lie in a shared low-loss basin (Wortsman et al., 2022). More generally, merging can be viewed as *interpolation* between checkpoints, where linear interpolation (LERP) produces models with smoothly varying behavior. Motivated by geometric considerations, practitioners also use spherical linear interpolation (SLERP) to interpolate on a hypersphere, which can better preserve norms and sometimes yields smoother blends than Euclidean averaging, especially when mixing two models (Goddard et al., 2024).

Another widely studied family of methods represents fine-tuning updates as task vector. In task arithmetic, a task vector is computed as the weight difference between a fine-tuned checkpoint and its base model, and multiple vectors can be added or scaled to compose behaviors (Ilharco et al., 2022). This view has driven many merging recipes for foundation models, including recent LLM-focused work that formalizes coefficient selection or scaling strategies for GPT-scale merges (Zhou et al., 2024; Zheng et al., 2024). However, naive task-vector addition is brittle: different experts can propose conflicting parameter updates, and fine-tuning noise accumulates in small-magnitude directions. To mitigate this, sparsification and conflict-resolution methods such as *TIES-Merging* trim low-salience updates, elect a consensus sign, and merge only sign-consistent parameters (Yadav et al., 2023). *DARE* further introduces stochastic pruning (drop-and-rescale) before merging, motivated by redundancy in fine-tuning deltas and the desire to suppress spurious components (Yu et al., 2024). Beyond global heuristics, recent methods attempt to localize where task knowledge resides, merging only small identified parameter regions (e.g., Localize-and-Stitch) to reduce interference and preserve pretraining knowledge (He et al., 2024).

A complementary line explores *data-aware* merging objectives that weight parameters by uncertainty or functional similarity. Fisher-weighted averaging treats merging as approximate Bayesian model combination using Fisher information as a curvature-aware metric (Matena & Raffel, 2022). Related approaches such as RegMean and dataless knowledge fusion estimate merge coefficients by minimizing prediction discrepancies across models, improving cross-domain generalization without full retraining (Jin et al., 2023). Others learn merge weights automatically (e.g., layer-wise or task-wise) using surrogate objectives such as entropy minimization on unlabeled data, aiming to reduce manual hyperparameter tuning (Yang et al., 2023). Finally, the rapid growth of merging techniques has motivated surveys and standardized evaluations; recent benchmarks such as MergeBench broaden coverage across instruction following, math, multilingual, coding, and safety, highlighting that merging quality depends strongly on base-model strength, expert compatibility, and coefficient/density choices (Yang et al., 2024; He et al., 2025b).

Most existing evaluations emphasize task accuracy, forgetting, or aggregate utility. Only recently have works begun to stress that merging can also propagate undesired behaviors—e.g., safety misalignment can transfer from a single poorly aligned expert into the merged model (Hammoud et al., 2024). In our setting, this motivates going beyond scalar reward or benchmark scores to examine *open-ended generation stability*: even when interpolation/extrapolation improves preference metrics, merges can amplify brittle or collapse-like behaviors (repetition, code-switching, nonsense), which are not well-captured by standard merging benchmarks.

# E    Additional Robustness Experiments

In this section, we report additional analyses to verify that the observed reward–degeneration trade-off is robust to evaluator choice, degeneration threshold, and decoding configuration.

## E.1    Reward Model Robustness

Our main experiments use `weqweasdas/RM-Mistral-7B` as the reward model, following prior work. To assess evaluator dependence, we repeat the Tulu-7b-DPO extrapolation experiment (Section 4.2) using `sfairXC/FsfairX-LLaMA3-RM-v0.1` (ranked first on RewardBench Chat within the same parameter scale as of 02/14/2026). Table 3 shows that both reward models exhibit the same qualitative pattern: moderate extrapolation (+0.3, +0.5) improves reward score, while excessive scaling (+0.8) degrades performance. This confirms that the extrapolation trend is not an artifact of a specific reward model.

| Reward Model | Base | +0.3 | +0.5 | +0.8 |
|---|---|---|---|---|
| RM-Mistral-7B | 5.05 | 5.24 | 5.41 | 5.02 |
| FsfairX-LLaMA3-RM-v0.1 | -1.47 | -1.25 | -1.07 | -1.27 |

Table 3: Reward score comparison under two reward models for Tulu-7b-DPO extrapolation.

## E.2    Judge Model Sensitivity

We evaluate degeneration using three judge models: GPT-4, GPT-4.1, and Gemini-Flash-2.5. Table 4 reports degeneration rates for the Tulu-7b-DPO extrapolation setting. Across all judges, degeneration increases monotonically with extrapolation coefficient. Although absolute percentages differ due to judge sensitivity, the relative ordering remains consistent.

| Judge Model | Base | +0.3 | +0.5 | +0.8 |
|---|---|---|---|---|
| GPT-4 | 33% | 40% | 45% | 52% |
| GPT-4.1 | 50% | 61% | 64% | 94% |
| Gemini-Flash-2.5 | 49% | 51% | 57% | 90% |

Table 4: Degeneration rate under different judge models (Top-100 threshold).

Additionally, we manually inspected the first 20 GPT-4 annotations and found 14/20 aligned with human judgment, indicating reasonable agreement for relative trend analysis.

## E.3    Degeneration Threshold Sensitivity

Our main metric selects the 100 lowest-reward responses. To test robustness, we vary the tail threshold (Top-50, 100, 150, 200). Table 5 shows degeneration rates using GPT-4.1 as judge.

While absolute values decrease as the threshold widens, the monotonic degeneration increase with larger extrapolation coefficients remains intact.

| Threshold | Base | +0.3 | +0.5 | +0.8 |
|---|---|---|---|---|
| Top-50 | 62% | 72% | 76% | 98% |
| Top-100 | 50% | 61% | 64% | 94% |
| Top-150 | 40.67% | 50% | 56.67% | 87.33% |
| Top-200 | 36.50% | 45.50% | 49.75% | 79% |

Table 5: Degeneration rate under different tail thresholds (GPT-4.1 judge).

### E.4 Decoding Robustness

We evaluate degeneration under multiple temperature settings for the +0.5 extrapolation configuration of Tulu-7b-DPO.

| Temperature | 0.0 | 0.3 | 0.7 | 1.0 |
|---|---|---|---|---|
| Degeneration Rate | 0.67 | 0.69 | 0.64 | 0.70 |

Table 6: Degeneration rate under different decoding temperatures (GPT-4.1 judge, Top-100 threshold).

Degeneration remains elevated across all temperatures, including greedy decoding ($T = 0.0$). This indicates that the instability is not merely a sampling artifact but reflects parameter-space effects induced by merging.

