# OpenReview forum: "The Pitfalls of Text Degeneration when Aligning LLMs through Model Merge"
_TMLR — Accepted by TMLR_

### Review · Reviewer_So5p · 2026-01-20

**Summary Of Contributions:**

This paper shows that although model merging is a cheap alternative to alignment training, merging decoder-only LLMs can silently induce model collapse, causing degenerate behaviors like repetition and nonsense even when reward scores improve, and demonstrates through large-scale experiments that existing merging methods (including sparsification-based ones) do not reliably prevent this, with stability depending strongly on how similar the merged models’ training histories and domains are.

**Audience:**

Yes

**Audience Explanation:**

Yes, model merging gains increasing attention recently in the field. This is an important topic.

**Broader Impact Concerns:**

NA.

**Claims And Evidence:**

No

**Claims Explanation:**

See requested changes below. The most important critique is about the misalignment between the paper's motivation and paper's experiments.

**Requested Changes:**

## Related Work
1. Please cite Wise-FT (Wortsman, Mitchell, et al. "Robust fine-tuning of zero-shot models." Proceedings of the IEEE/CVF conference on computer vision and pattern recognition. 2022.) properly in your introduction. I think this is more related than Task Arithmetic paper because Wise-FT interpolated EXACTLY 2 models, pretrained & fine-tuned, and they saw the interpolated model achieved the best trade-off between in-distribution and out-of-distribution performance. Task Arithmetic paper (addition for merging) focused more on multitask capability, rather than the best 2 end point metric tradeoff. Merging SFT & RL checkpoints and your motivation is closer to Wise-FT.
2. Besides, in the context of multitask model merging, decoder-based LLM went through different post-training processes ( or example SFT vs. GRPO, one RL setup not discussed), and is studied comprehensively in [1]. Advanced methods like [2,3] are also based on sparsification and can achieve "less forgetting on pretrained capability". I think this is highly related to your concern about collapse-like behavior (see comment bullet point 2 as well)

[1] He, Yifei, et al. "MergeBench: A Benchmark for Merging Domain-Specialized LLMs." arXiv preprint arXiv:2505.10833 (2025).

[2] He, Yifei, et al. "Localize-and-stitch: Efficient model merging via sparse task arithmetic." arXiv preprint arXiv:2408.13656 (2024).

[3] Wang, Ke, et al. "Localizing task information for improved model merging and compression." arXiv preprint arXiv:2405.07813 (2024).


## Comments
1. Your Sec. 4.4 experiment (2) and (5) to merge two models with different bases doesn't make sense. Model interpolation is only known to work with shared trajectories [1,2,3] theoretically and empirically due to linear mode connectivity theory.

[1] Frankle, Jonathan, et al. "Linear mode connectivity and the lottery ticket hypothesis." International Conference on Machine Learning. PMLR, 2020.

[2] Zhou, Zhanpeng, et al. "On the Emergence of Cross-Task Linearity in the Pretraining-Finetuning Paradigm." arXiv preprint arXiv:2402.03660 (2024).

[3] Zeng, Siqi, et al. "Task Vector Bases: A Unified and Scalable Framework for Compressed Task Arithmetic."

2. For all figures, report all metrics at weight = 0 and 1 to show the full trajectory. It's not surprising that the degeneration metric is in between the value at 0 and 1, and not worse than these 2 endpoints. That said, the behavior that this paper observed is normal, because one model that you try to merge in already had bad degradation performance. We do not expect to FURTHER suppress collapse behavior.
- Related to this point, since in your introduction you said that these collapse behavior ~= pretrained behavior loss. I expect you to merge post-trained models with pretrained base models as a baseline, and see if merging with sparsification based merging method can help.
- This is an important point as it's related to your motivation. I didn't get why do you expect model will not show collapse behavior when you interpolate between SFT and RL models.
3. In Sec 4.1, please provide full reward model details (which checkpoint do you use, how is the reward model trained etc) because that is directly related to your key metric throughout the paper.
4. I can't find llama3-8b-sft and llama3-8b-iter checkpoints on huggingface. This requires clarification from the authors.
5. GPT-4 judge prompts are missing. Judge accuracy (alignment with human ground truth labels is also missing).
6. Please use \citep and \citet properly.

---

> ### Author Response · Authors · 2026-02-17
> **Response to Reviewer So5p (1/2)**
>
> Dear Reviewer So5p,
>
> First and foremost, thank you for taking the time from your busy schedule to thoroughly read our paper, acknowledging its strengths, and providing valuable suggestions that have helped improve its quality. Following your insightful feedback, we have discussed and deliberated on each of your questions and comments. In response, we have tried our best to provide detailed explanations and conduct additional experiments. We look forward to your favorable consideration of our work, and wish you all the best and continued success in your endeavors.
>
> ---
>
> > **(Requested Changes: Related Work) Wise-FT and Additional Merging Literature.**
>
> **R:** We thank the reviewer for highlighting Wise-FT and the recent LLM merging literature.
> 1. We agree that Wise-FT is closely connected in spirit: it ensembles/interpolates weights between a zero-shot pretrained model (e.g., CLIP) and its fine-tuned counterpart to improve robustness to distribution shifts without additional training compute. **However, our setting differs in ways that matter for degeneration concerns. ** Specifically, WiSE-FT focuses on discriminative prediction (primarily CLIP-style models) and accuracy/robustness trade-offs under distribution shift. In contrast, we study **decoder-only generative LLMs** under **post-training alignment** (SFT/RL variants), where failure often manifests as degenerated open-ended generation rather than pure accuracy drops. Besides, Wise-FT evaluates in-/out-of-distribution accuracy. It does not target degeneration phenomena that can appear even when performance metrics improve. We will cite WiSE-FT in the introduction and use it to better motivate why interpolation can look good on aggregate metrics while still hiding stability failures in generative settings.
>
> 2. MergeBench provides a large-scale evaluation suite for merging domain-specialized LLMs, emphasizing multi-domain performance, forgetting, and practical guidance for algorithm selection. [2] proposes identifying highly sparse parameter regions that encode task-relevant updates and “stitching” them back into a base model to reduce interference and preserve pretrained knowledge. [3] argues that task information may remain present post-merge but is not properly utilized due to interference. We agree that these works are highly related. Our key distinction is not that they are “about capability trade-offs” (they do more than that), but that **their primary evaluations center on task performance/robustness/forgetting**, whereas **our paper targets a different failure axis: collapse-like degeneration in open-ended generation**. We will revise the related work to add a detailed discussion of these works.
>
> ---
>
> >**(Comments 1) Model interpolation is only known to work with shared trajectories due to linear mode connectivity.**
>
> **R:** We agree that linear mode connectivity theory predicts that interpolation works best for models sharing training trajectories. However, our Section 4.4 explicitly includes cross-base merging **to test robustness under realistic but imperfect practitioner usage**. In practice, users frequently merge models from different bases. Public merged models on leaderboards often ignore trajectory compatibility. Our empirical finding is precisely aligned with linear mode connectivity theory: 1) Same-base interpolation is more stable. 2) Cross-base interpolation is unstable and may degrade performance. Thus, these experiments are not meaningless. They validate theoretical expectations and demonstrate practical risks. We will clarify in the revision.
>
> ---
>
> > **(Comment 2) For all figures, report all metrics at weight = 0 and 1 to show the full trajectory. It is not surprising degeneration lies between endpoints. One model already had bad degeneration; we do not expect further suppression.**
>
> **R:** We apologize that the figures might cause confusion and will improve the captions. However, **both endpoints are actually included in all relevant experiments**:
> * Figure 4 (Extrapolation): includes λ = 0 (original RL model) and λ = 0.8 (aggressive extrapolation), following prior work [1].
> * Figure 5 (Interpolation): explicitly shows “Model A” and “Model B” as the two endpoints.
> * Figure 6 (TIES/Dare-TIES): endpoints correspond to the green and red lines.
> * Figure 7 (Expert integration): coefficient = 0 represents the original expert models (endpoint).
>
> Our key claim is **not** that degeneration should be suppressed below both endpoints. Rather, our central finding is that **reward scores can increase while degeneration simultaneously increases**. This is not a trivial convex interpolation effect. We will clarify this in the revision.
>
> ---

---

> ### Author Response · Authors · 2026-02-17
> **Response to Reviewer So5p (2/2)**
>
> > **(Comment 2a) Since collapse ~= pretrained capability loss, merge post-trained models with pretrained base.**
>
> **R:** We appreciate this suggestion. However, we do not claim that these collapse behavior ~= pretrained behavior loss. Pretrained base models typically exhibit high diversity and broad coverage, **but lack instruction-following alignment**.  Our study focuses on the degeneration that emerges after alignment-driven merging, and does not correspond to reverting to base-model behavior. Collapse-like degeneration is not merely “undoing alignment” or recovering pretrained diversity. It reflects instability introduced by combining post-training updates. Therefore, merging pretrained base models is a reasonable additional baseline. **However, it is orthogonal to our main focus**, which is to analyze stability trade-offs when combining post-training alignment updates (e.g., SFT and RL deltas). We will clarify this distinction more explicitly in the revision.
>
> ---
> > **(Comment 2b) This is an important point as it's related to your motivation. I didn't get why do you expect model will not show collapse behavior when you interpolate between SFT and RL models.**
>
> **R:** We would like to clarify that we do not assume or expect interpolation to eliminate or promote collapse-like behavior. Our work aims to examine existing merging practices such as ExPO[1], which is often justified by improvements in reward or performance metrics. We investigate whether those improvements reliably reflect stable generative behavior. Our findings show that reward gains can coincide with increased degeneration, revealing a stability trade-off that is not obvious from alignment metrics alone. We will clarify this motivation more explicitly in the revision.
>
> ---
>
> > **(Comment 3) Reward model details.**
>
> **R:** As described in Section 4.1, we adopt a strong open-source reward model (weqweasdas/RM-Mistral-7B) following prior work[1]. **Our goal is not to claim absolute alignment quality, but to analyze relative trends under controlled merging settings**. The reward model serves as a **consistent axis of comparison** across merge coefficients and strategies. To further strengthen this point, we conducted additional experiments on the Tulu-7b-DPO extrapolation setting (Section 4.2), replacing the reward model with **sfairXC/FsfairX-LLaMA3-RM-v0.1** (**ranked first** on RewardBench Chat within the same parameter scale as of 02/14/2026). **The results exhibit the same qualitative trend**. Reward scores increase with moderate extrapolation (e.g., +0.3, +0.5) and decrease when the scaling coefficient becomes too large (+0.8). This confirms that the extrapolation–instability trade-off is not specific to a single reward model.
>
> | Reward Model     | Tulu-7b-dpo  | +0.3  | +0.5 | +0.8 |
> |-------------------|--------|--------|--------|------------|
> |weqweasdas/RM-Mistral-7B|5.05|5.24|5.41|5.02 |
> |FsfairX-LLaMA3-RM-v0.1        | -1.47 | -1.25 |  -1.07 | -1.27|
>
> ---
> > **(Comment 4) I can't find llama3-8b-sft and llama3-8b-iter checkpoints on huggingface.**
>
> Actually, we provide the open-source models and HuggingFace IDs in Appendix A. Table 2 shows that:
> | Model Name     | HuggingFace IDs  |
> |-------------------|--------|
> |llama3-8b-sft   |RLHFlow/LLaMA3-SFT|
> |llama3-8b-iter    |RLHFlow/LLaMA3-iterative-DPO-final|
>
> > **(Comment 5) GPT-4 judge prompts are missing. Judge accuracy.**
>
> **R:** We thank the reviewer for this insightful comment. In the revision, we will include the exact GPT-4 prompt used for degeneration labeling, along with detailed instructions provided to the judge model. Our objective is not absolute scoring, but to **examine relative degeneration trends** across merging configurations. We manually inspected the first 20 GPT-4 annotations and found 14/20 consistent with human judgment, suggesting reasonable reliability. To further address this concern, we evaluated degeneration using two additional judge models (GPT-4.1 and Gemini-Flash-2.5). **The trend remains consistent across judges**. Degeneration rates increase as the extrapolation coefficient grows.
>
> | Judge Model      | Tulu-7b-dpo | +0.3   | +0.5   | +0.8   |
> | ---------------- | ----------- | ------ | ------ | ------ |
> | GPT-4            | 33.00%      | 40.00% | 45.00% | 52.00% |
> | GPT-4.1          | 50.00%      | 61.00% | 64.00% | 94.00% |
> | Gemini-Flash-2.5 | 49.00%      | 51.00% | 57.00% | 90.00% |
>
>
>
> > **(Comment 6) Please use \citep and \citet properly.**
>
>
> We will correct all citation formatting.
>
> ---
> Reference:
> [1] Zheng, Chujie, et al. Weak-to-strong extrapolation expedites alignment.

---

### Review · Reviewer_i9u7 · 2026-02-01

**Summary Of Contributions:**

This paper investigates the risks of model merging (Task Arithmetic, TIES-Merging, Dare-TIES) when applied to decoder-based LLMs (post-trained). The experiments show that model merging can cause "text degeneration" and "model collapse" while standard performance metrics may improve. In particular,

- The authors propose a framework for identifying three types of text degeneration: repetition, code-switching, and nonsense. This framework complements standard reward model scores and provides insights into "model collapse".
- The authors list several interesting observations:
  - Extrapolation (of task vectors) may improve alignment but increases degeneration rates.
  - Complex merging methods still fail to prevent degeneration
  - Interpolation mostly yields in-between performance.
  - Merging is more stable when models start from the same base checkpoint.

**Audience:**

Yes

**Audience Explanation:**

This paper touches upon the timely topic of model merging, which addresses a broad audience in ML working with LLMs and alignment techniques. The resulting safety-related findings are relevant because they reveal that standard performance metrics can be misleading, showing improvements while models actually degrade through text degeneration. Additionally, the systematic documentation of when and why merging fails provides important negative results that can save researchers compute and time, and prevent deployment of unstable models.

**Broader Impact Concerns:**

This work provides important signals to practitioners about model merging. I believe it could prevent unstable models from being deployed but also the negative results sound too discouraging. for further research. I'd try to rephrase the conclusions from a more fruitful angle.

**Claims And Evidence:**

Yes

**Claims Explanation:**

Overall,
- The hypotheses are tested on 10+ models, ranging from 7B to 70B parameters, and using three well-known merging methods. The authors incude extrapolation, interpolation, and expert integration, making the conclusions more reliable.
- The three metrics (repetition, code-switching, nonsense) are explained well.
- The findings show consistent patterns across model sizes and architectures. Figures 4-7 reveal that these patterns do not stem from specific evaluation choices but are truly general phenomena.

**Requested Changes:**

Requested Changes
1. This sentence "To fill this gap, we study the failure modes of model merging for alignment and to understand when merging behaves..." sounds wrong.
2. A clear and formal definition of "model collapse" should be provided already in page 2 when you first introduce the concept, before Section 3. Additionally, I am unsure if "collapse" is the right term for what you observe, since the models still produce sensible outputs most of the time and only show degeneration in a subset of cases. I'd consider using more precise terminology like "partial degeneration" or "localized quality degradation".
3. The examples in Figure 2 are far worse than what's defined as model collapse (losing support for the tail, low-diversity outputs). These are clear mistakes and formatting failures, not subtle distribution shifts. I'd be happy if you can provide examples that better align with the definition of collapse, or clarify why you focus on these clearly mistaken cases rather than the definition, e.g., showing low diversity.
4. You rely on GPT-4 as a judge for labeling degeneration, but do not provide any evidence for the reliability of it. A human evaluation where GPT-4 labels are evaluated by human annotators is needed.
5. Is there a related work section that I'm missing? Please add this section and clearly position your contribution/novelties.
6. All experiments use fixed decoding parameters. What happens when different decoding parameters are used, e.g., smaller temperatures that might reduce degeneration? A comparison across at least 2-3 temperature settings (e.g., 0.3, 0.7, 1.0) is needed to understand if the findings are robust to these choices or better decoding is actually sufficient.
7. Why is the degeneration rate computed from exactly "100 responses with the lowest reward score"? Justification and comparisons with other sampling strategies (e.g., random sampling across all reward scores) are needed to show how robust the metric is.

---

> ### Author Response · Authors · 2026-02-17
> **Response to Reviewer i9u7 (1/2)**
>
> Dear Reviewer i9u7,
>
> First and foremost, thank you for taking the time from your busy schedule to thoroughly read our paper, acknowledging its strengths, and providing valuable suggestions that have helped improve its quality. We look forward to your favorable consideration of our work, and wish you all the best and continued success in your endeavors.
>
> ---
>
> > **(Requested Changes 1) This sentence "To fill this gap, we study the failure modes of model merging for alignment and to understand when merging behaves..." sounds wrong.**
>
> **R:** Thank you for pointing out the phrasing issue. To clarify, this sentence aims to express our goal of understanding when model merging acts as a benign capability composition versus when it causes instability. We systematically examine this within the context of alignment merging settings. In the revision, we will rewrite it to enhance clarity and directness.
>
> ---
>
> > **(Requested Changes 2) A clear and formal definition of "model collapse" should be provided already in page 2 when you first introduce the concept...**
>
> **R:** We thank the reviewer for this thoughtful and important suggestion. In the current draft, a formal definition of model collapse is provided in Section 3, which distinguishes collapse from related phenomena such as alignment tax and surface-level text degeneration. However, we agree that introducing the concept more precisely at its first appearance in Section 1 would substantially improve clarity and prevent potential misunderstanding. In the revision, we will clearly state that, in prior literature, model collapse refers to a training-time degenerative process—typically arising from recursive self-training on synthetic data. In our work, we observe collapse-like behavioral symptoms such as repetition and code-switching. Our intention is to highlight that merging can push models toward regimes exhibiting the same surface-level degenerative behaviors identified in collapse studies.
>
> We also appreciate the reviewer’s concern that the term “collapse” may overstate the phenomenon we observe, since the models still generate sensible outputs in many cases and degeneration manifests primarily in a subset of responses. In the revision, we will adopt more careful phrasing, such as “collapse-like behavior,” “partial degeneration,” or “localized quality degradation,” when describing our empirical findings.
>
> ---
>
> > **(Requested Changes 3) The examples in Figure 2 are far worse than what's defined as model collapse...**
>
> **R:** We thank the reviewer for this insightful comment. We agree that the examples shown in Figure 2 represent easily identifiable failures, such as repetition loops and emoji spam, rather than the distributional shrinkage described in formal definitions of model collapse. **Our intention in presenting these examples was diagnostic rather than definitional.** Specifically, we focus on observable surface-level manifestations of instability that are relevant in alignment settings. We observe **collapse-like behavioral symptoms** such as repetition and code-switching. We aim to highlight that **merging can push models toward regimes exhibiting the same surface-level degenerative behaviors** identified in collapse studies. We agree that the connection between our qualitative examples and the formal collapse definition can be articulated more clearly. In the revision, we will clarify that Figure 2 illustrates behavioral symptoms of instability, not the full spectrum of collapse phenomena.
>
> ---
>
> > **(Requested Changes 4) You rely on GPT-4 as a judge for labeling degeneration, but do not provide any evidence for the reliability of it.**
>
> **R:** We thank the reviewer for this insightful comment. Our objective is not absolute scoring, but to **examine relative degeneration trends** across merging configurations. We manually inspected the first 20 GPT-4 annotations and found 14/20 consistent with human judgment, suggesting reasonable reliability. To further address this concern, we evaluated degeneration using two additional judge models (GPT-4.1 and Gemini-Flash-2.5). **The trend remains consistent across judges**. Degeneration rates increase as the extrapolation coefficient grows.
>
> | Judge Model      | Tulu-7b-dpo | +0.3   | +0.5   | +0.8   |
> | ---------------- | ----------- | ------ | ------ | ------ |
> | GPT-4            | 33.00%      | 40.00% | 45.00% | 52.00% |
> | GPT-4.1          | 50.00%      | 61.00% | 64.00% | 94.00% |
> | Gemini-Flash-2.5 | 49.00%      | 51.00% | 57.00% | 90.00% |
>
>
> ---
>
> > **(Requested Changes 5) Is there a related work section that I'm missing?**
>
> **R:** Thank you for raising this. Currently, related work is integrated across Sections 1,2, and 3 (e.g., model collapse, alignment tax, degeneration literature, merging methods). However, we agree that separating it into a “Related Work” section would improve clarity. We will add a dedicated section that organizes prior work into model merging methods and text degeneration.

---

> ### Author Response · Authors · 2026-02-17
> **Response to Reviewer i9u7 (2/2)**
>
> > **(Requested Changes 6) All experiments use fixed decoding parameters.**
>
> **R:** We thank the reviewer for this constructive suggestion. First, we use the default decoding configuration aligned with ExPO [1], which empirically demonstrates merging effectiveness under standard top-k/nucleus sampling. **Our goal is to remain consistent with established merging evaluation protocols rather than tune decoding to favor particular outcomes.** Second, we conducted additional experiments on the Tulu-7b-DPO extrapolation setting (coefficient = 0.5, Section 4.2) under different temperature settings, using GPT-4.1 as the judge model.
>
> | Temperature       | 0.0  | 0.3  | 0.7  | 1.0  |
> | ----------------- | ---- | ---- | ---- | ---- |
> | Degeneration Rate | 0.67 | 0.69 | 0.64 | 0.70 |
>
> As shown above, degeneration rates remain consistently elevated across temperatures, including greedy decoding. While minor fluctuations are expected due to stochasticity, there is **no qualitative change in behavior**. The degeneration–extrapolation trade-off persists, indicating that the observed instability is not an artifact of a specific sampling temperature. We will include these results in the revision.
>
>
> ---
>
> > **(Requested Changes 7) Why is the degeneration rate computed from exactly "100 responses with the lowest reward score"?**
>
>
> **R:** Figure 3 shows that degenerated responses are concentrated in low reward-score bins. Degeneration is rare in high-score regions. Therefore, selecting the worst-scoring responses can isolate the region where severe failure manifests. Our degeneration metric is thus **not intended as a global quality score**, but as a **diagnostic for collapse-like failure modes**. Running degeneration labeling over the entire dataset would dilute the signal, since high-quality generations rarely exhibit degeneration. We agree that robustness to threshold choice is important. We re-evaluated degeneration rates under multiple tail thresholds in the Tulu-7B-DPO extrapolation setting (Section 4.2), using GPT-4.1 as the judge model.
>
> | Threshold | Tulu-7b-dpo | +0.3   | +0.5   | +0.8   |
> | --------- | ----------- | ------ | ------ | ------ |
> | Top-50    | 62.00%      | 72.00% | 76.00% | 98.00% |
> | Top-100   | 50.00%      | 61.00% | 64.00% | 94.00% |
> | Top-150   | 40.67%      | 50.00% | 56.67% | 87.33% |
> | Top-200   | 36.50%      | 45.50% | 49.75% | 79.00% |
>
> Across all thresholds, the key qualitative pattern remains unchanged: **increasing the extrapolation coefficient consistently leads to higher degeneration rates**. While the absolute percentages decrease as the threshold widens (since less severe samples are included), the relative ordering is preserved. This robustness across thresholds reinforces our central claim.
>
> ---
>
> Reference:
> [1] Zheng, Chujie, et al. Weak-to-strong extrapolation expedites alignment.

---

### Review · Reviewer_pnW9 · 2026-02-09

**Summary Of Contributions:**

This paper investigates the failure modes of model merging when applied to align decoder-based LLMs with human preferences. The authors examine two merging scenarios: (1) extrapolation of post-training task vectors (SFT -> RL delta added back to RL weights) and (2) interpolation between pairs of SFT/RL models from shared or different base checkpoints. The study covers three merging methods (Task Arithmetic, TIES-Merging, Dare-TIES) across open-source models ranging from 7B to 70B parameters. The authors score all responses with a reward model to assess text degeneration, select the 100 lowest-scoring responses and annotate them by GPT-4 for three degeneration types: repetition, code-switching and nonsense. The core finding is that model merging can induce text degeneration and collapse-like behavior even when aggregate reward scores appear to improve and that advanced sparsification-based merging methods do not reliably prevent this degeneration. The authors also show that merging models with shared training pipelines or domain characteristics is empirically safer.

Strengths: The paper addresses a practically important problem by providing preliminary experiments across model scales and families, introduces a taxonomy of text degeneration types and reveals an interesting asymmetry between domain experts (code vs. math) in merging stability.

Weaknesses: The evaluation is limited and narrow (single reward model + LLM judge on a tail subset), the extrapolation is conceptually questionable, critical baselines and ablations are missing.

**Additional Comments:**

The paper provides preliminary experiments that reward model scores can improve while generation quality degrades and this is definitely of practical value. However, the current evidence does not convincingly establish this claim due to the limited evaluation metrics and missing ablations/baselines.

The motivation for model merging over multi-task training is insufficiently justified. Multi-task SFT is also cheaper and more effective for combining capabilities. I think it is important for the authors to motivate this more clearly and explain why people should care about this problem in the manuscript.

I think the paper would benefit more from proposing a practical pre-merge diagnostic (e.g., task vector similarity or weight-space distance) that can be predictive of when degeneration is likely rather than only reporting post-hoc failures.

Some questions for the authors:
1. What is the exact GPT-4 prompt used for degeneration annotation? How sensitive is the degeneration rate to the choice of judge model?
2. Why were only 100 worst responses chosen for analysis? How do results change with different thresholds?
3. Have you experimented with adding deltas to different reference models rather than the same RL model in the extrapolation setup?
4. Could the math vs. code asymmetry be explained by formatting differences in training data rather than fundamental domain incompatibility?

**Audience:**

Yes

**Audience Explanation:**

Model merging is useful as a low-cost method for combining model capabilities and many merged models appear on public leaderboards without systematic quality analysis. The core question of whether model merging can silently degrade generation quality even when aggregate metrics improve is practically important and might be relevant to the community. The preliminary findings about the disconnect between reward model scores and actual generation quality and findings about domain compatibility for merging would be useful. However, the findings for some of the experiments are expecgted and it would be beneficial to make the work more thorough and comprehensive.

**Broader Impact Concerns:**

No ethical concerns

**Claims And Evidence:**

No

**Claims Explanation:**

While the paper presents preliminary experimental results, the evidence does not convincingly support several of its central claims:

1. The entire degeneration analysis relies on a single reward model to select the worst 100 responses, followed by GPT-4 annotation. A single reward model introduces systematic bias, and the "degeneration rate" is computed over a cherry-picked tail subset (worst 100 by reward score), making it unclear how representative this metric is of overall model behavior. The authors do not report results over any other complementary metrics such as Self-BLEU for lexical diversity, perplexity under a reference model for fluency, n-gram repetition rate as a model-free repetition detector and human evaluation on a subset of random samples. I think it is insufficient to only rely on a single scalar metric.

2. The paper provides no information about the exact GPT-4 prompt used for annotation, sensitivity to different judge models or how borderline cases are handled. The phrase "introduce no new information" (used to define degenerate repetition in Section 3.1) is vague. What constitutes "new information" and how GPT-4 is anchored for this is unclear.

3. Computing delta = RL weights - SFT weights and adding this back to the RL model is equivalent to RL + lambda * (RL - SFT), which simply amplifies RL-specific updates. I think it is expected that aggressive amplification along an already-optimized RL model might degrade quality. The RL model has moved away from SFT, so linearly extrapolating in weight space may push into undesired regions. A more principled setup would compute domain-specific deltas and add them to a general-purpose RL model. The paper does not discuss why this particular extrapolation direction is sensible or compare to alternatives.

4. The paper frames it as a finding that "retaining influential parameters" does not prevent degeneration, but this is predictable: parameters with the largest magnitude changes are precisely those most aggressively modified during RL, so retaining them preferentially retains features driving both alignment and degeneration. A more principled approach would identify parameters influential in the *target* model and regularize those during merging to prevent existing capabilities from large updates. The paper does not explore this direction.

5. The term "model collapse" [1] refers specifically to recursive self-consuming training loops. The phenomenon observed here in this work is text degeneration from weight-space interpolation/extrapolation and I think it can be better described as "misalignment amplification" or "parameter space instability." Honestly, I'm not sure if "model collapse" is a broader term with a different interpretation and can be misleading.

6. Several baselines and ablations are missing. No comparison against multi-task SFT or joint training to assess whether degeneration is specific to merging or inherent to combining heterogeneous capabilities. No ablation on the degeneration threshold (why 100 worst responses?). No experiments with different decoding strategies to test whether degeneration is an artifact of the specific sampling configuration.

References:

[1] Ilia Shumailov, Zakhar Shumaylov, Yiren Zhao, Nicolas Papernot, Ross Anderson, and Yarin Gal. AI models collapse when trained on recursively generated data.

**Requested Changes:**

From high to low priority:

1. The reliance on a single reward model + GPT-4 judge is the most significant limitation. The authors should report additional metrics such as Self-BLEU for lexical diversity, n-gram repetition rate and perplexity under a held-out reference model to name a few. These metrics should be easy to compute over the full response set, not just the worst 100 samples.

2. The authors should include details about the prompts used for GPT-4 annotation for text degeneration analysis and also conduct a human evaluation on a sample of reasonable size to measure agreement with GPT-4. Additionally, the authors should also perform a sensitivity analysis of choosing a different judge model in the results.

3. Report degeneration rates when selecting the worst 50, worst 200, and full response set instead of only the worst 100. Show that the observed trends are robust to this choice.

4. Include at least one baseline where models are jointly fine-tuned on the combined data (e.g., joint SFT on chat + math data, or chat + code data) for Section 4.5. This will be useful to understand whether degeneration is specific to merging or inherent to combining heterogeneous capabilities and also addresses the practical motivation question of why one would merge rather than jointly train if the computational cost is cheaper.

5. Add MBPP scores for Chat+Math and GSM8K scores for Chat+Code. This reveals whether adding one domain expert actively harms performance in another domain, which is more useful than only showing target-domain improvement alongside general reward decrease.

6. Report the results for different merge strategies under greedy decoding and different sampling configurations to verify that degeneration patterns are not artifacts of the sampling setup. All current experiments use a single configuration.

7. Discuss why adding delta = (RL - SFT) back onto the RL model is a sensible choice, given that it simply amplifies the RL optimized model. Compare with alternatives where domain-specific delta is added to a general-purpose RL model.

8. Explain why magnitude-based parameter selection conflates beneficial and harmful features in the discussion part of the section in the manuscript. Consider an alternative where parameters influential in the target model's existing capabilities are identified and regularized during merging, rather than retaining the largest delta values.

9. Math training data typically contains chain-of-thought reasoning, which should preserve instruction-following abilities. Investigate whether the degeneration stems from formatting differences (e.g., LaTeX, structured answer patterns) conflicting with the chat template. Break down degeneration by type (repetition vs. nonsense vs. code-switching) separately for each expert combination.

10. The authors should provide a detailed analysis of whether degeneration severity changes across the model scale of the same family and discuss implications. Tulu series models might be a natural choice for this analysis.

11. Use `\citep` for parenthetical citations (e.g., "ExPO Zheng et al. (2024)" → "ExPO (Zheng et al., 2024)"). Fix the citation format across the manuscript.

12. The authors should also report results grouped by model family in addition to the current results to understand if the observations are consistent across the model families or if model family-specific.

---

> ### Author Response · Authors · 2026-02-17
> **Response to Reviewer pnW9 (1/4)**
>
> Dear Reviewer pnW9,
>
> First and foremost, thank you for taking the time from your busy schedule to thoroughly read our paper, acknowledging its strengths, and providing valuable suggestions that have helped improve its quality. Following your insightful feedback, we have discussed and deliberated on each of your questions and comments. In response, we have tried our best to provide detailed explanations and conduct additional experiments. We look forward to your favorable consideration of our work, and wish you all the best and continued success in your endeavors.
>
> ---
>
> > **1. (Weakness 5) On the Use of the Term “Model Collapse.”**
> >“The term ‘model collapse’ refers specifically to recursive self-consuming training loops…”
>
> **R:** We appreciate this important clarification. In Shumailov et al., “model collapse” is indeed studied in the context of recursive self-training on synthetic data. However, the **observable consequences** of collapse in that line of work include reduced diversity, increased repetition, incoherence, and low-quality outputs. In our work, **we do not claim that merging induces collapse via recursive data contamination**. Rather, we observe **collapse-like behavioral symptoms** such as repetition and code-switching. Our intention is to highlight that **merging can push models toward regimes exhibiting the same surface-level degenerative behaviors** identified in collapse studies, even though the underlying mechanism differs (weight-space extrapolation/interpolation rather than recursive training). We agree that terminology clarity is important. In the revision, we will clarify that we study **collapse-like degeneration** rather than recursive collapse in the strict sense.
>
> ---
> > **2. (Weakness 1; Requested Changes 1, 2 & 3) Reliance on a Single Reward Model + GPT-4 Judge.**
> > “The entire degeneration analysis relies on a single reward model… cherry-picked tail subset… insufficient to rely on a single scalar metric.”
>
>
> **R:** We appreciate the reviewer’s concerns regarding evaluation robustness.
>
> **2a)** On the use of a single reward model.
>
> We respectfully disagree that our evaluation constitutes cherry-picking. As described in Section 4.1, we adopt a strong open-source reward model (weqweasdas/RM-Mistral-7B) following prior work[1]. **Our goal is not to claim absolute alignment quality, but to analyze relative trends under controlled merging settings**. The reward model serves as a **consistent axis of comparison** across merge coefficients and strategies. To further strengthen this point, we conducted additional experiments on the Tulu-7b-DPO extrapolation setting (Section 4.2), replacing the reward model with **sfairXC/FsfairX-LLaMA3-RM-v0.1** (ranked first on RewardBench Chat within the same parameter scale as of 02/14/2026). **The results exhibit the same qualitative trend**. Reward scores increase with moderate extrapolation (e.g., +0.3, +0.5) and decrease when the scaling coefficient becomes too large (+0.8). This confirms that the extrapolation–instability trade-off is not specific to a single reward model.
>
> | Reward Model     | Tulu-7b-dpo  | +0.3  | +0.5 | +0.8 |
> |-------------------|--------|--------|--------|------------|
> |weqweasdas/RM-Mistral-7B|5.05|5.24|5.41|5.02 |
> |FsfairX-LLaMA3-RM-v0.1        | -1.47 | -1.25 |  -1.07 | -1.27|
>
>
> **2b)** On GPT-4 as judge and judge sensitivity
>
> We agree that different judge models may introduce bias [2]. Our objective is again not absolute scoring, but to **examine relative degeneration trends** across merging configurations. We manually inspected the first 20 GPT-4 annotations and found 14/20 consistent with human judgment, suggesting reasonable reliability. To further address this concern, we evaluated degeneration using two additional judge models (GPT-4.1 and Gemini-Flash-2.5). **The trend remains consistent across judges**. Degeneration rates increase as the extrapolation coefficient grows.
>
> | Judge Model      | Tulu-7b-dpo | +0.3   | +0.5   | +0.8   |
> | ---------------- | ----------- | ------ | ------ | ------ |
> | GPT-4            | 33.00%      | 40.00% | 45.00% | 52.00% |
> | GPT-4.1          | 50.00%      | 61.00% | 64.00% | 94.00% |
> | Gemini-Flash-2.5 | 49.00%      | 51.00% | 57.00% | 90.00% |
>
>
>
>
>
> **2c)** On additional automatic metrics
>
>
> We agree that automatic metrics such as Self-BLEU, perplexity, and repetition statistics are important for evaluating overall generation quality. These metrics provide useful signals about diversity, fluency, and surface-level repetition, and are widely adopted in text generation evaluation. However, **they are orthogonal to the primary focus of our paper**, which is to diagnose collapse-like degeneration. For example, Self-BLEU measures diversity across outputs but does not directly capture code-switching or incoherent/nonsensical segments. Similarly, n-gram repetition statistics may fail to reflect degeneration when repetition is not the dominant failure mode.

---

> ### Author Response · Authors · 2026-02-17
> **Response to Reviewer pnW9 (2/4)**
>
> > **3. (Weakness 2; Requested Change 2) Labeling Prompt Details.**
> “No information about the exact GPT-4 prompt... the phrase ‘introduce no new information’ is vague.”
>
> **R:** In the revision, we will include the exact GPT-4 prompt used for degeneration labeling, along with detailed instructions provided to the judge model. We will also clarify the definition of “introduce no new information” in the context of repetition. Specifically, repetition will be defined as token-, phrase-, or sentence-level recurrence that does not contribute additional semantic content or advance the response relative to the instruction[3][4]. Importantly, humans may repeat phrases for emphasis, clarification, or rhetorical structure. We only classify repetition as degenerate when it exhibits clear self-looping behavior or redundant restatement without semantic progression. For example, the following output would be labeled as degenerate repetition:
>
> ```
> The age of the crater is about 3 . 6 billion years and it has been in the
> proximity of the south lunar pole for at least 10,000 years . The South
> Crater is located on the southern edge of the northern highlands . The South
> Crater is located on the southern edge of the northern highlands . The South
> Crater is located on the southern edge of the northern highlands . …
> ```
>
> ---
>
> > **4. (Weakness 1; Requested Change 3) Robustness to Different Threshold Choice.**
> > “Why only 100 worst responses?”
>
> **R:** Figure 3 shows that degenerated responses are concentrated in low reward-score bins. Degeneration is rare in high-score regions. Therefore, selecting the worst-scoring responses can isolate the region where severe failure manifests. Our degeneration metric is thus **not intended as a global quality score**, but as a **diagnostic for collapse-like failure modes**. Running degeneration labeling over the entire dataset would dilute the signal, since high-quality generations rarely exhibit degeneration. We agree that robustness to threshold choice is important. We re-evaluated degeneration rates under multiple thresholds in the Tulu-7B-DPO extrapolation setting (Section 4.2), using GPT-4.1 as the judge model.
>
> | Threshold | Tulu-7b-dpo | +0.3   | +0.5   | +0.8   |
> | --------- | ----------- | ------ | ------ | ------ |
> | Top-50    | 62.00%      | 72.00% | 76.00% | 98.00% |
> | Top-100   | 50.00%      | 61.00% | 64.00% | 94.00% |
> | Top-150   | 40.67%      | 50.00% | 56.67% | 87.33% |
> | Top-200   | 36.50%      | 45.50% | 49.75% | 79.00% |
>
> Across all thresholds, the key qualitative pattern remains unchanged: **increasing the extrapolation coefficient consistently leads to higher degeneration rates**. While the absolute percentages decrease as the threshold widens (since less severe samples are included), the relative ordering is preserved. This robustness across thresholds reinforces our central claim.
>
>
> ---
>
> > **5. (Weakness 6; Requested Change 4) Why Not Include Joint Multi-Task Fine-Tuning?**
> > “No comparison against multi-task SFT or joint training…”
>
> **R:** We agree that joint SFT is a meaningful baseline. However, our focus is **not to compare merging against optimal training pipelines**, but to investigate the *failure modes of merging itself*, which is widely used as a low-cost, data-free alternative. Joint multi-task SFT requires access to combined data, retraining compute, and does not reflect the common practical scenario of post-hoc merging of released checkpoints. **Our goal is to answer when practitioners merge aligned models to save cost, what risks might arise.**
>
> ---
>
> > **6. (Requested Change 5) Cross-Domain Metrics.**
> > “Add MBPP for Chat+Math and GSM8K for Chat+Code...”
>
> **R:** In Figure 7, we **already demonstrate that incorporating a domain expert can negatively affect performance outside its target domain**, revealing a clear cross-domain trade-off. The primary goal of Section 4.5 is to analyze the relationship between domain specialization, reward alignment, and degeneration, rather than to comprehensively benchmark every cross-domain combination. In particular, the math expert (MetaMath) is trained specifically to improve mathematical reasoning, not coding ability. **Therefore, evaluating MBPP for Chat+Math would be expected to yield poor performance, offering limited additional diagnostic value.** Similarly, the coding expert is not optimized for mathematical reasoning, so GSM8K performance for Chat+Code is unlikely to meaningfully illuminate the degeneration mechanism. More importantly, **our central finding in this section is not that cross-domain accuracy drops—which is largely predictable—but that domain gains are tightly coupled with increases in degeneration and decreases in reward scores.** This instability is the phenomenon we aim to highlight.

---

> ### Author Response · Authors · 2026-02-17
> **Response to Reviewer pnW9 (3/4)**
>
> > **7. (Requested Change 6) Decoding Sensitivity.**
>
> **R:** We thank the reviewer for this constructive suggestion. First, we use the default decoding configuration aligned with ExPO [1], which empirically demonstrates merging effectiveness under standard top-k/nucleus sampling. **Our goal is to remain consistent with established merging evaluation protocols rather than tune decoding to favor particular outcomes.** Second, we conducted additional experiments on the Tulu-7b-DPO extrapolation setting (coefficient = 0.5, Section 4.2) under different temperature settings, using GPT-4.1 as the judge model.
>
> | Temperature       | 0.0  | 0.3  | 0.7  | 1.0  |
> | ----------------- | ---- | ---- | ---- | ---- |
> | Degeneration Rate | 0.67 | 0.69 | 0.64 | 0.70 |
>
> As shown above, degeneration rates remain consistently elevated across temperatures, including greedy decoding. While minor fluctuations are expected due to stochasticity, there is **no qualitative change in behavior**. The degeneration–extrapolation trade-off persists, indicating that the observed instability is not an artifact of a specific sampling temperature. We will include these results in the revision.
>
> ---
>
> > **8. (Weakness 3; Requested Change 7) Extrapolation Setup: Why Add (RL − SFT) Back onto RL?**
> > “Computing delta = RL weights − SFT weights and adding this back to the RL model… simply amplifies RL-specific updates… The paper does not discuss why this particular extrapolation direction is sensible or compare to alternatives.”
>
> **R:** We follow the ExPO framework[1], which has **already empirically demonstrated gains from this extrapolation direction**. Our contribution is not to propose a novel extrapolation direction, but to show that even **empirically validated merging strategies can silently increase degeneration risk**. The paper's core message is diagnostic: even when reward improves under established merging strategies, degeneration may increase. We will clarify this positioning more explicitly.
>
>
> ---
>
> > **9. (Weakness 4; Requested Change 8) Magnitude-Based Parameter Selection.**
> > “The paper frames it as a finding that ‘retaining influential parameters’ does not prevent degeneration, but this is predictable…”
>
> **R:** We agree, and this is precisely one of our empirical findings. Section 4.3 shows that TIES and Dare-TIES sparsification preserve reward score but fail to reliably reduce degeneration. This suggests alignment-beneficial and degeneration-driving features are intertwined in ($\Delta\theta$). **Magnitude heuristics are insufficient to disentangle them.** Identifying which subsets of parameters encode “helpful” versus “harmful” post-training effects is intrinsically challenging. Post-training parameter change ($\Delta\theta$) is highly entangled across layers and modules, and alignment procedures such as DPO/RLHF modify representations in a distributed manner rather than through isolated, easily separable subnetworks. We agree that developing principled, interpretable mechanisms for separating these effects is an important research direction. However, to the best of our knowledge, current model merging practice remains largely black-box and heuristic-driven. While we acknowledge this as a promising and important direction for the community, **it is orthogonal to the primary goal of our work, which is diagnostic: we aim to systematically characterize the failure modes of existing merging techniques rather than propose a new mechanistic disentanglement method.**

---

> ### Author Response · Authors · 2026-02-17
> **Response to Reviewer pnW9 (4/4)**
>
> > **10. (Requested Change 9) Does Math CoT or Formatting Cause Degeneration?**
> > “Could degeneration stem from formatting differences...”
>
> **R:** We respectfully disagree that formatting alone explains the observed instability. The degeneration types observed (repetition loops, emoji spam, nonsense) are not merely formatting mismatches. The degeneration rate rises smoothly with coefficient scaling, indicating parameter-space amplification rather than template mismatch. **If formatting conflict were the main driver, we would expect immediate instability at small coefficients.**
>
> ---
>
> > **11. (Requested Change 10) Degeneration Across Model Scale.**
> > “The authors should provide a detailed analysis of whether degeneration severity changes across the model scale of the same family and discuss implications…”
>
> **R:** We agree that analyzing how degeneration varies across model scale is an important direction. In Section 4.2, we include experiments on the Tulu-2 series (7B, 13B, and 70B). Within the extrapolation setting studied there, **we do not observe a clear trend indicating that larger models are systematically more or less robust to degeneration.** The reward–degeneration trade-off appears to persist across scales, suggesting that the phenomenon is not confined to small models. We acknowledge that model behavior may vary under different merging settings. A more comprehensive scale analysis would require systematically sweeping merging strategies, densities, coefficients, and model families while controlling for other variables. **Conducting such an extensive study would significantly broaden the scope of the paper beyond its primary focus, which is to diagnose collapse-like degeneration under representative alignment-oriented merging setups.** We will explicitly acknowledge this as a limitation in the revised manuscript and highlight systematic scale analysis of merge stability as an important direction for future work.
>
> ---
>
>
> > **12. (Requested Change 11) Citation Format.**
>
> **R:** We will correct citation formatting and use \citep consistently in the revision.
>
> ---
>
> > **13. (Requested Change 12) Group Results by Model Family.**
>
> **R:** We respectfully note that we already perform model-family-level analysis in the current manuscript. First, in Section 4.2, we evaluate extrapolation across multiple model families, including Zephyr-7B-alpha, Starling-7B-alpha, Llama3-8B-iter, and the Tulu-2 series (7B/13B/70B). These models originate from different base checkpoints (e.g., Mistral, LLaMA2, LLaMA3) and distinct post-training pipelines. The reward–degeneration trade-off is shown consistently across these families (Figure 4), indicating that the phenomenon is not confined to a single architecture. Second, Section 4.4 explicitly analyzes interpolation under same-base vs. different-base families, including SFT+SFT and RL+RL combinations across shared and mismatched checkpoints. The RL+RL-3 (cross-base) instability case directly addresses model-family mismatch effects. These results demonstrate that merging behavior depends strongly on whether models share a common base or post-training trajectory.
>
> ---
>
> Reference:
>
> [1] Zheng, Chujie, et al. Weak-to-strong extrapolation expedites alignment.
>
> [2] Shi, Lin, et al. Judging the judges: A systematic study of position bias in llm-as-a-judge
>
> [3] Li, et al. Repetition in repetition out: Towards understanding neural text degeneration from the data perspective.
>
> [4] Xu, et al. Learning to break the loop: Analyzing and mitigating repetitions for neural text generation.

---

### Author Response · Authors · 2026-02-26
**Summary of Revisions**

Dear Reviewers,

First and foremost, thank you for taking the time from your busy schedule to thoroughly read our paper, acknowledging its strengths, and providing valuable suggestions that have helped improve its quality. Following your insightful feedback, we have discussed and deliberated on each of your questions and comments. In response, we have tried our best to provide detailed explanations and additional experiments. In summary, we revised the paper in three main ways:

* **Terminology clarification:** We clarify that our work studies *collapse-like / localized text degeneration* (e.g., repetition, code-switching, nonsense) rather than recursive “model collapse” in the strict sense, and we adjust the wording throughout to avoid ambiguity.

* **Stronger robustness evaluation:** We add ablations demonstrating that the reward–degeneration trend remains consistent across (i) a second reward model, (ii) multiple judge models (GPT-4, GPT-4.1, Gemini-Flash-2.5), (iii) different tail thresholds (Top-50/100/150/200), and (iv) different decoding temperatures, including greedy decoding. We also include a small human agreement check on judge labels and provide the full degeneration-annotation prompt for transparency.

* **Related work and formatting improvements:** We add a dedicated Related Work section (including WiSE-FT, MergeBench, and Localize-and-Stitch) and correct citation formatting (consistent use of \citep/\citet).

We look forward to your favorable consideration of our work and wish you continued success in your endeavors!

Best,

The Authors

---

### Decision · Action_Editor_LqZg · 2026-04-02

**Recommendation:** Accept as is

**Audience:**

Yes

**Audience Explanation:**

The results and insights offered help deepen our understanding of failure cases in model merging. The observations and findings could be of great interest to the model merging community.

**Claims And Evidence:**

Yes

**Claims Explanation:**

This paper presents an empirical study of text degeneration in model merging. Through experiments on multiple LLMs and three merging methods, the authors show that text degeneration can emerge even when reward scores improve.

The paper addresses a timely and practical problem. The results and insights offered help deepen our understanding of failure cases in model merging. The experimental setup is sound, the scope is reasonable, and the observations and findings could be of great interest to the model merging community.

The main limitation is that the paper remains largely descriptive. It demonstrates empirically that text degeneration occurs but offers limited insight into why it happens or how to overcome it. Reviewers also raised concerns about the relatively low GPT-4/human agreement rate, the absence of a joint training baseline, the lack of evaluation with larger reward models, and the restriction of failure cases to three specific types of text degeneration.

Overall, this paper provides a useful empirical contribution supported by sound experimental results, which meets the standard for TMLR.